# Clinical, laboratory and radiological characteristics and outcomes of novel coronavirus (SARS-CoV-2) infection in humans: A systematic review and series of meta-analyses

Israel Júnior Borges do Nascimento[1,2◉]*, Thilo Caspar von Groote[3◉], Dónal P. O'Mathúna[4,5], Hebatullah Mohamed Abdulazeem[6], Catherine Henderson[7], Umesh Jayarajah[8], Ishanka Weerasekara[9,10], Tina Poklepovic Pericic[11], Henning Edgar Gerald Klapproth[12], Livia Puljak[13], Nensi Cacic[11], Irena Zakarija-Grkovic[11], Silvana Mangeon Meirelles Guimarães[1], Alvaro Nagib Atallah[14], Nicola Luigi Bragazzi[15], Milena Soriano Marcolino[1], Ana Marusic[11], Ana Jeroncic[11]*, On behalf of the International Task Force Network of Coronavirus Disease 2019 (InterNetCOVID-19)[¶]

1 University Hospital and School of Medicine, Universidade Federal de Minas Gerais, Belo Horizonte, Minas Gerais, Brazil, 2 Medical College of Wisconsin, Milwaukee, Wisconsin, United States of America, 3 Department of Anaesthesiology, Intensive Care and Pain Medicine, University Hospital Münster, Münster, Germany, 4 Helene Fuld Health Trust National Institute for Evidence-based Practice in Nursing and Healthcare, College of Nursing, The Ohio State University, Columbus, Ohio, United States of America, 5 School of Nursing, Psychotherapy and Community Health, Dublin City University, Dublin, Ireland, 6 Department of Sport and Health Sciences, Technische Universität München, Munich, Germany, 7 Swanscoe Communications Ltd, Macclesfield, United Kingdom, 8 Department of Surgery, Faculty of Medicine, University of Colombo, Colombo, Sri Lanka, 9 Department of Physiotherapy, Faculty of Allied Health Sciences, University of Peradeniya, Peradeniya, Sri Lanka, 10 School of Health Sciences, Faculty of Health and Medicine, The University of Newcastle, Callaghan, Australia, 11 Cochrane Croatia, University of Split School of Medicine, Split, Croatia, 12 Department of Internal Medicine D, University Hospital Münster, Münster, Germany, 13 Center for Evidence-Based Medicine, Catholic University of Croatia, Zagreb, Croatia, 14 Cochrane Brazil; Evidence-Based Health Program, Universidade Federal de São Paulo, São Paulo, Brazil, 15 Laboratory for Industrial and Applied Mathematics (LIAM), Department of Mathematics and Statistics, York University, Toronto, Ontario, Canada

◉ These authors contributed equally to this work.
¶ Membership of InterNetCOVID-19 is provided in the Supplementary Appendix (S1 File).
* israeljbn@ufmg.br and israeljrbn@gmail.com (IJBN); ajeronci@mefst.hr (AJ)

## Abstract

New evidence on the COVID-19 pandemic is being published daily. Ongoing high-quality assessment of this literature is therefore needed to enable clinical practice to be evidence-based. This review builds on a previous scoping review and aimed to identify associations between disease severity and various clinical, laboratory and radiological characteristics. We searched MEDLINE, CENTRAL, EMBASE, Scopus and LILACS for studies published between January 1, 2019 and March 22, 2020. Clinical studies including ≥10 patients with confirmed COVID-19 of any study design were eligible. Two investigators independently extracted data and assessed risk of bias. A quality effects model was used for the meta-analyses. Subgroup analysis and meta-regression identified sources of heterogeneity. For hospitalized patients, studies were ordered by overall disease severity of each population

**Data Availability Statement:** All relevant data are within the manuscript and its Supporting Information files.

**Funding:** CH is employed by Swanscoe Communications Ltd, a medical writing company, and has previously provided medical writing support for medical and pharmaceutical projects, unrelated to the current work. Medical writing for the current work was provided by CH pro bono and was not funded by any pharmaceutical or device company.

**Competing interests:** We have read the journal's policy and the authors of this manuscript have the following potential competing interests: CH is employed by Swanscoe Communications Ltd, a medical writing company, and has previously provided medical writing support for medical and pharmaceutical projects, unrelated to the current work. CH has previously provided medical writing support for projects funded by Amgen, AstraZeneca, Novartis, Sobi and Takeda, unrelated to the current work. This does not alter our adherence to PLOS ONE policies on conflicts of interest. All other authors declare that no competing interests exist.

and this order was used as the modifier variable in meta-regression. Overall, 86 studies (n = 91,621) contributed data to the meta-analyses. Severe disease was strongly associated with fever, cough, dyspnea, pneumonia, any computed tomography findings, any ground glass opacity, lymphocytopenia, elevated C-reactive protein, elevated alanine aminotransferase, elevated aspartate aminotransferase, older age and male sex. These variables typically increased in prevalence by 30–73% from mild/early disease through to moderate/severe disease. Among hospitalized patients, 30–78% of heterogeneity was explained by severity of disease. Elevated white blood cell count was strongly associated with more severe disease among moderate/severe hospitalized patients. Elevated lymphocytes, low platelets, interleukin-6, erythrocyte sedimentation rate and D-dimers showed potential associations, while fatigue, gastrointestinal symptoms, consolidation and septal thickening showed non-linear association patterns. Headache and sore throat were associated with the presence of disease, but not with more severe disease. In COVID-19, more severe disease is strongly associated with several clinical, laboratory and radiological characteristics. Symptoms and other variables in early/mild disease appear non-specific and highly heterogeneous.

**Clinical Trial Registration:** PROSPERO CRD42020170623.

## Introduction

In December 2019, the appearance of a novel coronavirus caused an outbreak of respiratory infections originating from the Huanan food market in Wuhan, China [1, 2], which has since developed into a global pandemic. The novel coronavirus is referred to as "severe acute respiratory syndrome coronavirus 2" (SARS-CoV-2) potentially resulting in "coronavirus disease" (COVID-19) [3]. As of August 8, 2020, 19,497,292 patients have been infected globally, and 723,854 deaths have been recorded [4]. Clinical symptoms of infection with SARS-CoV-2 are heterogeneous and vary in severity; some patients may even be asymptomatic [5, 6]. In some patients, COVID-19 leads to serious outcomes, such as acute respiratory distress syndrome, coagulation dysfunction and death [5, 7, 8]. However, many questions about the clinical syndrome of COVID-19 remain unanswered.

The evidence base on COVID-19 is rapidly expanding. The large volume of data and short timeframe challenges clinicians, researchers and policy makers worldwide. Comprehensive, systematic reviews of the available literature are needed in order to effectively summarize evidence. In addition, as would be expected in the early stages of a pandemic involving a novel virus, much of the data is derived from observational case series, providing only limited evidence. Publication timescales are exceptionally short given the urgency of the situation and balancing the need for rapid dissemination of information with established reporting standards is challenging. Rigorous assessment of study methodology and outcomes is therefore especially important in order to facilitate evidence-based clinical practice and policy.

Our research group recently performed a scoping review and meta-analysis that analyzed data published up until February 24, 2020 [5]. The work reported here substantially builds on this using the same stringent protocol to encompass subsequent studies. The aim was to systematically analyze all clinical evidence on the COVID-19 pandemic published in the peer-reviewed literature. We aimed to look at the impact of severity of disease on clinical, laboratory

and radiological characteristics, and outcomes and specifically investigated differences between the overall COVID-19 patient group and patients with severe disease.

## Methods

### Data sources and information searches

This review follows meta-analysis of observational studies in epidemiology (MOOSE) guidelines [9] and is reported in accordance with the preferred reporting items for systematic reviews and meta-analyses (PRISMA) [10]. The review was registered in PROSPERO (CRD42020170623). We restricted the sample size in eligible studies to at least 10 patients, as the minimum to calculate a valid percentage. There were two minor deviations from the original protocol: in quantitative synthesis we excluded studies in which all patients or the majority of patients were children, and we introduced subgroup analysis based on disease severity.

We searched the following five databases: MEDLINE, CENTRAL, EMBASE, Scopus, and LILACS from January 1, 2019 to March 22, 2020. The searches had no restrictions on language or publication format. The search strategy (Supplementary S1 Appendix in S1 File) was designed and conducted in collaboration with an experienced information specialist. Reference lists of relevant studies were screened to identify additional publications.

### Study selection

Studies eligible for inclusion reported confirmed cases of SARS-CoV-2 in humans, including at least 10 patients in the study sample, and were published in a peer reviewed journal. *In vitro* and mathematical modelling studies were excluded. Studies were included irrespective of the clinical study design or publication language. COVID-19 was defined as a diagnosis by any specific test such as positive SARS-CoV-2 PCR, genetic sequence analysis or IgM/IgA antibody detected in a serological assay. Two independent investigators conducted the title/abstract screening and full-text screening. The study selection was performed using the Covidence platform. Articles deemed potentially eligible during title/abstract screening were retrieved as full text for further screening. In both stages of screening (titles/abstracts and full texts), discrepancies were resolved by a third review author. Articles published in languages other than English were translated by a native/fluent speaker.

### Data collecting process and quality assessment

Two investigators independently extracted data using customized tables and were blinded to each other until data extraction was finished. Discrepancies were then resolved via consensus between a group of senior researchers in the team. Data were carefully assessed for any overlap of study populations. If any overlap was suspected, an assessment was made based on hospitals involved and the time frame of the study. In cases of overlapping, the most recent study was included.

The primary outcomes were all-cause mortality, and prevalence of clinical symptoms, laboratory findings and chest imaging findings in COVID-19. Other data analyzed included demographic characteristics, comorbidities, incubation period, treatment provided, pharmacotherapy, admission to the intensive care unit (ICU), length of stay in the ICU and in hospital. Data were included from baseline to follow-up. If studies reported multiple follow-up time points, most recent data was analyzed. Data from severely ill patients were comparable in terms of participant characteristics and outcome measures, and therefore pooled together in a meta-analysis.

Two researchers independently assessed the risk of bias of case series and epidemiological studies using the 'Methodological Quality and Synthesis of Case Series and Case Reports Protocol' [11], derived from the Newcastle-Ottawa Scale (NOS), and randomized clinical trials were assessed using the Cochrane risk of bias tool for randomized trials (RoB) [12]. For the first tool, two questions were not applicable to our systematic review, and therefore excluded: "Was there a challenge/rechallenge phenomenon?" and "Was there a dose-response effect?" [13]. Disagreements were resolved by consulting a third reviewer.

## Data analysis

For dichotomous data, we extracted data for the number of events and total number of patients in order to perform proportion meta-analysis. For continuous data we extracted means and standard deviations. If data were presented as medians, and interquartile ranges or ranges, we estimated means and standard deviations using the method described by Wan et al. [14].

Meta-analyses of clinical, radiological and laboratory data, as well as data on clinical management and epidemiological characteristics of included patients were conducted using MetaXL v5.2 (EpiGear International, Sunrise Beach, Australia). To perform meta-analysis of proportions, data were transformed by double arcsine transformation and normalized [15]. Pooled proportions or means and 95% confidence intervals (CI) were calculated using the quality effects model (QE) [16]. The method was chosen over the random effects model because of high heterogeneity of the data. QE stabilizes the variance and mitigates the issue of a conventional random effects model underestimating standard errors in presence of high heterogeneity [17, 18]. The quality effects model also allows incorporation of information on study quality into the analysis. Its bias adjustment was found to be easily implementable with fewer limitations than the random effects model [19]. Heterogeneity was assessed by Cochrane's Q test considering a statistically significant value for $p < 0.1$, and Higgins $I^2$ [20].

Studies were assigned to subgroups based on the study sample. Studies enrolling initially asymptomatic patients that were then followed over time were categorized as '*initially asymptomatic*'. '*Early*' subgroup studies monitored onset of early symptoms of the disease in patients mainly identified through epidemiological tracking of close contacts. The study by Han et al. was also assigned to this category since the study investigated early clinical symptoms in patients who were admitted to hospital because of mild pneumonia [21]. The '*all-comers*' subgroup contained studies enrolling consecutive patients who had a positive PCR test, regardless of symptom presentation. These studies would be expected to be the most representative of COVID-19-positive individuals in the general population in our analysis because they included asymptomatic patients and outpatients as well as hospitalized patients. We included epidemiological reports that collected data from Centers for Disease Control, hospital laboratories nation/region-wide, and studies that collected data from multi/single-center hospital laboratories that also collected data from outpatients and asymptomatic patients. Finally, studies assigned to an '*admitted-to-hospital*' subgroup analyzed exclusively patients who were admitted to a hospital. Data on severely ill patients (defined as patients requiring care in an intensive care unit and/or requiring invasive or non-invasive ventilation) were extracted for further analysis.

We assessed statistical heterogeneity by examining the $I^2$, while also considering magnitude and direction of effects and strength of evidence. Heterogeneity was defined according to the Cochrane Handbook [22]; it might not be important heterogeneity (0–40%), moderate heterogeneity (30–60%), substantial heterogeneity (50–90%) and considerable heterogeneity (75–100%) [22]. Pooled point estimates are reported where $I^2 < 90\%$ and where we judged that estimates were not methodologically or clinically too diverse to be pooled; otherwise prevalence

ranges are reported. Methodological and clinical heterogeneity of studies, in particular sampling bias, was a significant source of heterogeneity. Patients were sampled at different stages of disease progression. In people identified through close contacts/epidemiological tracking usually only initial symptoms were observed ('early' studies). Some studies identified cases through laboratory testing, therefore including a wide spectrum of disease severities ('all-comer' studies). Other studies specifically included initially asymptomatic patients, pregnant women, or patients admitted to hospital (representative of more severe cases).

In order to investigate the association of various clinical, laboratory and radiological characteristics with severity of disease, we categorized hospital-based studies according to the overall disease severity of patients in each study and ordered studies by increasing severity within each category in forest plots. The same order of studies was used for all analyses. This severity index was determined according to the number of patients in each study meeting criteria from the 'Diagnosis and treatment protocol for novel coronavirus pneumonia', published by the China National Health Commission [23]. 'Mild' was defined as mild clinical symptoms with no sign of pneumonia on imaging; 'moderate' as fever and respiratory symptoms with radiological findings of pneumonia; 'severe' as meeting any of the following criteria: 1) respiratory distress ($\geq$30 breaths/min); 2) oxygen saturation $\leq$93% at rest; 3) arterial partial pressure of oxygen/fraction of inspired oxygen $\leq$300mmHg (1mmHg = 0.133kPa); 'critical': cases meeting any of the following criteria: a) respiratory failure and requiring mechanical ventilation; b) shock; c) with other organ failure requiring ICU care. As the QE model utilizes information on RoB and adjusts for bias, we did not perform sensitivity analysis for high- and low- quality studies. However, we performed one-out sensitivity analyses, where we excluded one study at a time and evaluated the impact of removing each of the studies on the summary results and the between-study heterogeneity.

## Results

### Study characteristics and risk of bias

The flow of studies through the search and selection process is presented in Fig 1. Overall 90 studies met the inclusion criteria (n = 92,620). Our previously published review included 60 studies [5]. Of these studies, 43 studies were not included in this update because our revised methodology included only studies with $\geq$10 patients. The main characteristics of the included studies are summarized in S1 Table in S1 File. Studies excluded because of overlapping are summarized in S2 Table in S1 File, and studies excluded for other reasons are shown in S3 Table in S1 File. A total of 86 studies were included in the meta-analyses (n = 91,621), and these were 29 case series, 45 consecutive retrospective case series, 2 consecutive prospective case series, a single RCT and 9 epidemiological reports. An additional 4 studies met the inclusion criteria, but the study populations were neonates or children, and these studies were excluded from the meta-analyses to reduce heterogeneity. Patients were from 13 countries (China, 92.7%; United States, 2.1%; South Korea, 2.1%; Singapore, 2.1%; and 9 countries across Europe, 1.0%). In most publications, COVID-19 was diagnosed according to the 'Diagnosis and treatment protocol for novel coronavirus pneumonia', published by the China National Health Commission [23]. Overall, included studies were judged as being of high risk of bias (S3–S5 Tables in S1 File; S1–S2 Figs in S1 File). The only included RCT had high risk of performance and detection bias, as the trial was not blinded; the risk of bias in the remaining five domains was low [24]. The majority of the studies were retrospective case series with risk of selection bias and selective reporting.

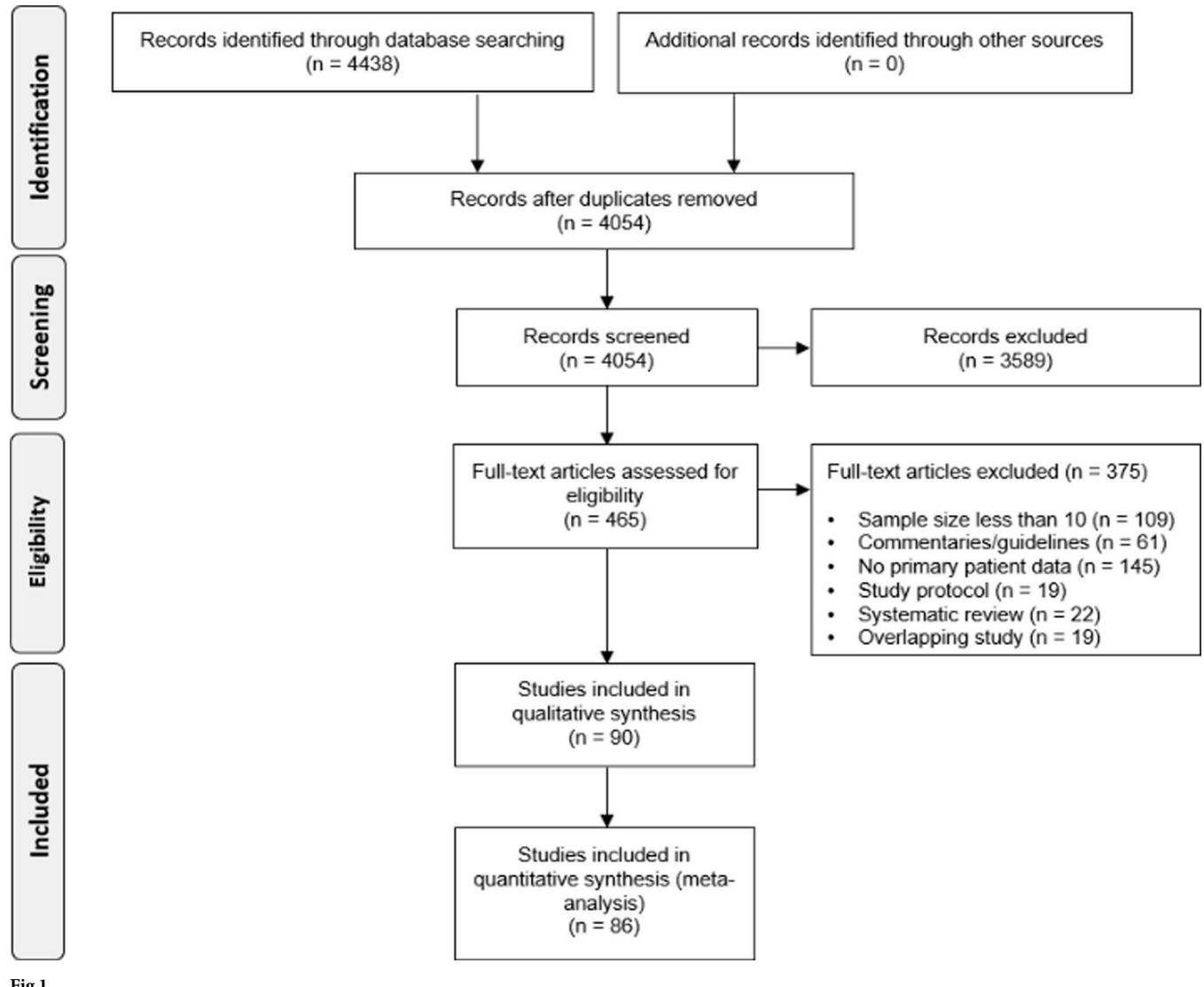

**Fig 1.**

## Overall associations

As summarized in Table 1, we identified five groups of variables: those associated with the presence of disease or worsening clinical status (14 variables), those with borderline associations (4 variables), those showing non-linear association patterns (4 variables), those with no apparent association (1 variable) and those with too few studies/too little data to indicate any association (6 variables).

## Patient age and sex

Findings from the quality effects models of mean age and prevalence of men among included studies are summarized in Table 2 and forest plots are presented in the Supplementary Appendix in S1 File. Severity of disease increased with age. Namely, patients admitted to hospital from the moderate/severe group were significantly older than patients from the mild/moderate group: 54 years [95% CI 51–57] vs. 42 [95% CI 38–46]. Moreover, within the moderate/severe

**Table 1. Summary of associations between variables and presence of disease or more severe disease states.**

| Variable[a] | Impact | Number of patients | Number of studies |
|---|---|---|---|
| *Variables associated with presence of disease or more severe disease states* | | | |
| **Age** | Strongly associated with more severe disease | 82,444 | 59 |
| **Male sex** | Strongly associated with more severe disease | 11,130 | 72 |
| **Fever** | Strongly associated with more severe disease | 7,658 | 61 |
| **Cough** | Strongly associated with more severe disease | 7,473 | 63 |
| **Dyspnea** | Strongly associated with more severe disease | 4,620 | 42 |
| **Any CT finding** | Strongly associated with more severe disease | 2,706 | 40 |
| **Any GGO** | Strongly associated with more severe disease | 1,239 | 20 |
| **Lymphocytes–low** | Strongly associated with more severe disease | 2,926 | 30 |
| **CRP–elevated** | Strongly associated with more severe disease | 2,305 | 26 |
| **ALT–elevated** | Strongly associated with more severe disease | 1,966 | 16 |
| **AST–elevated** | Strongly associated with more severe disease | 2,026 | 17 |
| **WBC–low** | Moderate trend for decreasing prevalence with more severe disease | 3,364 | 29 |
| **WBC–elevated** | Associated with presence of disease; strongly associated with more severe disease in moderate/severe patients admitted to hospital | 3,341 | 26 |
| **Pneumonia** | Associated with the presence of disease but no significant trend with more severe disease among patients admitted to hospital | 3,086 | 33 |
| **Sore throat** | Associated with the presence of disease, but not with more severe disease | 4,563 | 29 |
| **Headache** | Associated with the presence of disease, but not with more severe disease | 4,335 | 38 |
| *Variables with potential associations* | | | |
| **Lymphocytes–elevated** | Potential trend for decreasing prevalence with more severe disease | 506 | 6 |
| **Platelets–low** | Potential trend for decreasing prevalence with more severe disease in moderate/severe patients admitted to hospital | 1,829[b] | 12 |
| **IL-6 –elevated** | Potential strong association with more severe disease | 480 | 6 |
| **ESR–elevated** | Potential association with more severe disease | 647 | 9 |
| **D-dimer–elevated** | Potential association with more severe disease | 1,657 | 12 |
| *Variables showing non-linear association patterns* | | | |
| **Consolidation** | Two clusters of studies in moderate/severe patients admitted to hospital | 1,608 | 21 |
| **Septal thickening** | Two clusters of studies in moderate/severe patients admitted to hospital | 739 | 8 |
| **GI symptoms** | Delayed increasing trend in mild/moderate patients; decreasing trend in moderate/severe patients | 4,205 | 50 |
| **Fatigue** | Delayed increasing trend in mild/moderate patients; decreasing trend in moderate/severe patients | 5,942 | 50 |
| *No apparent association with presence of disease or worsening clinical status* | | | |
| **Platelets–elevated** | No association | 498 | 6 |
| *Too few studies / too little data to indicate any association* | | | |
| **Unilateral GGO** | N/A | 160 | 2 |
| **Neutrophils–elevated** | N/A | 811 | 11 |
| **Neutrophils–low** | N/A | 636 | 8 |
| **PCT–elevated** | N/A | 947 | 13 |
| **Troponin–elevated** | N/A | 181 | 3 |

[a]Laboratory variables were defined as the prevalence of high or low levels above or below specified thresholds, rather than as continuous variables.

[b]Includes the all-comer study by Guan WJ et al., 2020 [25] (n = 869).

group, the older age was associated with worse clinical status (Table 2, Source of heterogeneity). In severely ill patients, in 4 out of 5 studies the mean age was ≥54 years.

**Table 2. Age and sex findings from the quality effects models, across all studies and in subgroups.**

| | Overall | Initially asymptomatic | Early | Pregnant | All-comers | Admitted to hospital | | | Severely ill patients | Source of heterogeneity identified by meta-regression or subgroup analysis in hospitalized patients |
| | | | | | | All | Mild / moderate | Moderate / severe | | |
|---|---|---|---|---|---|---|---|---|---|---|
| **Age (mean years)** | 29.30–70.00 (n = 82,444; 59 studies) | 32.50–49.00 (n = 79; 2 studies) | 42.00–55.00 (n = 1,085; 8 studies) Pooled 47.81; 95% CI 42.55–53.07 Q = 21.52; *p<0.01*; I² = 72% | 29.30–32.00 (n = 103; 4 studies) Pooled 30.06; 95% CI 28.83–31.29 Q = 4.07; *p* = 0.25; I² = 26% | 42.60–52.60 (n = 77,448; 7 studies) | 35.00–70.00 (n = 3,729; 38 studies) | 35.00–52.80 (n = 1,735; 22 studies) | 41.60–70.00 (n = 1,994; 16 studies) | 45.50–70.00 (n = 192; 5 studies) | **All hospitalized patients:** Moderate/severe patients are older than mild/moderate 54 years, 95% CI 51.1–57.2 vs. 42 years, 95% CI 38.0–46.4 **Moderate/severe patients:** Older age with increasing severity R² = 29%; *p* = 0.032 |
| **Sex (% male)** | 0.22–0.76 (n = 11,130; 72 studies) | 0.33–0.40 (n = 79; 2 studies) Pooled 0.38; 95% CI 0.28–0.49 Q = 0.28; *p* = 0.60; I² = 0% | 0.22–0.76 (n = 629; 8 studies) Pooled 0.30; 95% CI 0.12–0.52 Q = 62.60; *p<0.01*; I² = 89% | – | 0.41–0.58 (n = 5,372; 9 studies) Pooled 0.53; 95% CI 0.49–0.56 Q = 27.87; *p<0.01*; I² = 71% | 0.33–0.76 (n = 5,050; 53 studies) | 0.33–0.76 (n = 2,515; 29 studies) Pooled 0.52; 95% CI 0.49–0.56 Q = 42.94; *p* = 0.04; I² = 35% | 0.38–0.67 (n = 2,535; 24 studies) Pooled 0.56; 95% CI 0.53–0.60 Q = 42.89; *p* = 0.01; I² = 46% | – | **Moderate/severe patients:** More men among patients with more severe clinical signs: 0.61 of men in studies with more (95%CI 0.58–0.65) vs. 0.53 in those with less (95%CI 0.49–0.56) severe clinical signs **Mild/moderate patients (only studies with more severe clinical signs):** Increasing prevalence of men with increasing severity R² = 60%; *p<0.001* |

CI, confidence interval.

Pooled prevalence estimates and 95% CIs are shown where statistical heterogeneity was below 90% (I²<90).

Regarding the sex distribution, in patients admitted to hospital male sex was associated with worse clinical status. In moderate/severe group there were significantly more men than women (pooled prevalence, 56% of men; 95% CI 53–60%). In addition, when we compared studies including more *vs* those including less severe patients within the moderate/severe group; we found that men were more often severely ill: 61% of men [95% CI 58–65%] vs. 53% [49–56%]. In mild/moderate hospitalized patents, the evidence on association of sex with clinical severity of COVID19 stemmed from studies reporting on more severely ill cases in which the prevalence of men increased with increasing COVID19 severity of included patients (Table 2, Source of heterogeneity). Contrary to hospitalized patients, 'initially asymptomatic' patients were more likely women: pooled prevalence, 38% of men [95%CI 28–49%]. Similar was found for 'early' studies, but due to considerable heterogeneity between studies, the evidence was uncertain.

## Symptoms

Table 3 shows the prevalence ranges of the most commonly reported symptoms, along with pooled prevalence estimates where study heterogeneity was acceptable; forest plots for each symptom are presented in the Supplementary Appendix in S1 File. Fever was extremely common among patients admitted to hospital (pooled prevalence 84%; 95% CI 80–87%). Regardless, it still showed strong association with patient's clinical status and a good predictive power for COVID19 outcomes. Specifically, moderate/severe patients presented with fever more commonly than mild/moderate: 89% [95% CI 86–92%] vs. 77% [72–82%]. Also, in hospitalized as well as in severely ill patents, the prevalence of fever increased with worsening of patients' clinical status (Table 3, Source of heterogeneity). Among two studies of initially asymptomatic patients, some of whom later became symptomatic, the pooled frequency of fever at some point during the course of infection was 16% (95% CI 6–29%) [26, 27]. Among 'early symptom' studies, the prevalence of fever was very heterogeneous ($p<0.01$; $I^2 = 99\%$).

Prevalence of cough appeared to increase from ~35% as estimated in 'early' and 'pregnant' studies to above 50% in patients admitted to hospital. However, due to high heterogeneity of studies this increasing trend was not statistically significant. Even among patients admitted to hospital, prevalence of cough varied considerably (2–92%). Most of this variability was assigned to studies enrolling moderate/severe patients, which formed two distinct groups of studies. One group included 10 studies tightly clustered together around the pooled prevalence of 79% (95% CI 77–82%) and showing no heterogeneity, whereas the second group of 9 studies was widely dispersed (2–63%). We found that cough was associated with the severity of disease in hospitalized patients, as moderate/severe patients from homogeneous cluster of studies reported cough more frequently than the mild/moderate group: 79%, 95% CI 77–82% vs. 54%, 47–61. In initially asymptomatic patients, 12% (95% CI 6–20%) developed cough during the course of the infection.

The prevalence of dyspnea across all studies is shown in Fig 2. Dyspnea was uncommon in all four studies of early disease (pooled prevalence 6%; 95% CI 2–11%). Among patients admitted to hospital, there was a wide variation in prevalence (1–81%). Nevertheless, dyspnea at admission was strongly associated with severity of disease in hospitalized patients, revealing its good predictive power. Notably, prevalence of dyspnea increased with severity of patients' clinical status in: mild/moderate and moderate/severe groups of studies, as well as in severely ill subgroup (Table 3, Source of heterogeneity). However, presence of dyspnea was not exclusive hallmark of severity of disease as even among the 14 studies of moderate/severe patients, there were six studies reporting a prevalence of 15% or less [31–35]. Among severely ill patients, the pooled prevalence of dyspnea was 51% (95% 39–64%), but there was also a report on zero occurrence of dyspnea among 13 severely ill patients as well [29].

Among hospitalized patients, similar patterns were found for both fatigue and gastrointestinal symptoms, with a trend for increasing prevalence with severity among mild/moderate patients and then decreasing prevalence with severity among moderate/severe, and severely ill patients (Table 3 and forest plots in the Supplementary Appendix in S1 File). Sore throat was not associated with the severity of disease. However, except for patients from 'initially asymptomatic' studies who didn't report any sore throat, this symptom was on average present in 8–12% of patients in various settings, from the 'early' studies to the severely ill subgroup. Similar was found for the headache. We found no association of headache with severity of disease, the symptom was present on average in 10–14% of patients from all the analyzed groups. The only exception were patients from 'initially asymptomatic' studies in whom the pooled prevalence of headache was significantly lower: 4%; 95 CI 1–7%.

Table 3. Frequency of mortality, symptoms, pneumonia and asymptomatic disease, from the quality effects models, across all studies and in subgroups.

| | Overall | Initially asymptomatic | Early | Pregnant | All-comers | Admitted to hospital | | | Severely ill patients | Source of heterogeneity identified by meta-regression or subgroup analysis in hospitalized patients |
| --- | --- | --- | --- | --- | --- | --- | --- | --- | --- | --- |
| | | | | | | All | Mild / moderate | Moderate / severe | | |
| **Mortality** | 0.00–0.067 (n = 58,542; 44 studies) | 0.00–0.00 (n = 79; 2 studies) Pooled 0.6% (95% CI 0–2.9%) Q = 0.08; $p = 0.78$; $I^2 = 0\%$ | 0.00–0.05 (n = 158; 4 studies) Pooled 0.023; 95% CI 0.00–0.06 Q = 3.62; $p = 0.31$; $I^2 = 17\%$ | 0.00–0.08 (n = 61; 4 studies) Pooled 0.026; 95% CI 0.00–0.07 Q = 1.81; $p = 0.61$; $I^2 = 0\%$ | 0.003–0.028 (n = 55,532; 9 studies) | 0.00–0.67 (n = 2,712; 25 studies) | 0.00–0.14 (n = 1,210; 10 studies) Pooled 0.01 (95% CI >0.001–0.042; Q = 102.77; $p<0.01$; $I^2 = 56\%$ | 0.00–0.67 (n = 1,502; 15 studies) | 0.00–1.00 (n = 803; 13 studies) | **Moderate/severe, and severely ill patients:** Increasing occurrence of death with increasing severity Moderate/severe: $R^2 = 73\%$; $p<0.001$ Severely ill patients: $R^2 = 66\%$; $p<0.001$ |
| **Fever** | 0.13–0.98 (n = 7,658; 61 studies) | 0.13–0.25 (n = 79; 2 studies) Pooled 0.16; 95% CI 0.06–0.29 Q = 1.75; $p = 0.19$; $I^2 = 43\%$ | 0.32–0.95 (n = 659; 7 studies) | 0.24–0.87 (n = 61; 4 studies) Pooled 0.58; 95% CI 0.26–0.87 Q = 16.81; $p<0.01$; $I^2 = 82\%$ | 0.44–0.91 (n = 3,169; 7 studies) | 0.61–0.98 (n = 3,690; 41 studies) Pooled 0.84; 95% CI 0.80–0.87 Q = 1.75; $p = 0.19$; $I^2 = 43\%$ | 0.61–0.96 (n = 1,305; 22 studies) Pooled 0.77; 95% CI 0.72–0.82 Q = 81.17; $p<0.01$; $I^2 = 73\%$ | 0.79–0.98 (n = 2,385; 19 studies) Pooled 0.89; 95% CI 0.86–0.92 Q = 53.01; $p<0.01$; $I^2 = 68\%$ | 0.47–1.00 (n = 1045; 13 studies) | **All hospitalized patients:** Fever is more prevalent in moderate/severe than mild/moderate 0.89; 95% CI 0.86–0.92 vs. 0.77; 95% CI 0.72–0.82 **All hospitalized, and severely ill patients:** Increasing prevalence with increasing severity All hospitalized patients: $R^2 = 17\%$; $p = 0.008$ Severely ill patients: $R^2 = 33\%$; $p = 0.041$ |
| **Cough** | 0.02–0.92 (n = 7,473; 63 studies) | 0.08–0.13 (n = 79; 2 studies) Pooled 0.12; 95% CI 0.06–0.20 Q = 0.20; $p = 0.65$; $I^2 = 0\%$ | 0.18–0.80 (n = 659; 7 studies) Pooled 0.36; 95% CI 0.16–0.58 Q = 51.06; $p<0.01$; $I^2 = 88\%$ | 0.15–0.60 (n = 61; 4 studies) Pooled 0.34; 95% CI 0.17–0.54 Q = 54.01; $p<0.01$; $I^2 = 56\%$ | 0.18–0.69 (n = 3187; 7 studies) | 0.02–0.92 (n = 3487; 43 studies) | 0.22–0.83 (n = 1226; 23 studies) Pooled 0.54; 95% CI 0.47–0.61 Q = 119.3; $p<0.001$; $I^2 = 81\%$ | 0.02–0.92 (n = 2261; 19 studies) Two clusters identified: Cluster 1 (n = 1006; 10 studies) Pooled 0.79; 95% CI 0.77–0.82 Q = 4.8; $p = 0.85$; $I^2 = 0\%$ Cluster 2 0.02–0.63 (n = 993; 9 studies) | 0.26–0.96 (n = 921; 14 studies) Two clusters identified: 0.77–0.96 0.26–0.54 | **All hospitalized patients (without cluster 2):** Cough is more prevalent in moderate/severe in mild/moderate patients 0.79; 95% CI 0.77–0.82 vs 0.54; 95% CI 0.47–0.61 |

(Continued)

Table 3. (Continued)

| | Overall | Initially asymptomatic | Early | Pregnant | All-comers | Admitted to hospital | | | Severely ill patients | Source of heterogeneity identified by meta-regression or subgroup analysis in hospitalized patients |
|---|---|---|---|---|---|---|---|---|---|---|
| | | | | | | All | Mild / moderate | Moderate / severe | | |
| **Fatigue** | 0.03–0.91 (n = 5,942; 50 studies) | 0.08 (n = 24; 1 study) | 0.04–0.39 (n = 618; 5 studies) | 0.19–0.31 (n = 44; 3 studies) Pooled 0.28; 95% CI 0.14–0.42 Q = 0.55; $p = 0.76$; $I^2 = 0\%$ | 0.07–0.44 (n = 2280; 6 studies) | 0.03–0.91 (n = 2976; 35 studies) | 0.03–0.91 (n = 1112; 18 studies) | 0.06–0.75 (n = 1864; 17 studies) | 0.10–0.83 (n = 836; 12 studies) Pooled 0.40; 95% CI 0.29–0.51 Q = 81.53; $I^2 = 86\%$ | **Mild/moderate:** Increasing prevalence with increasing severity $R^2 = 29\%$; $p = 0.018$ **Moderate/severe and severely ill:** Decreasing prevalence with increasing severity Moderate/severe patients: $R^2 = 23\%$; $p = 0.068$ (without 2 outliers) Severely ill patients: $R^2 = 29\%$; $p = 0.018$ |
| **Dyspnea** | 0.00–0.81 (n = 4620; 42 studies) | 0.00 (n = 24; 1 study) | 0.00–0.08 (n = 132; 4 studies) Pooled 0.06; 95% CI 0.02–0.11 Q = 0.83; $p = 0.84$; $I^2 = 0\%$ | 0.06–0.23 (n = 61; 4 studies) Pooled 0.13; 95% CI 0.05–0.22 Q = 2.00; $p = 0.57$; $I^2 = 0\%$ | 0.01–0.19 (n = 2145; 5 studies) | 0.01–0.81 (n = 2258; 28 studies) | 0.01–0.46 (n = 793; 14 studies) | 0.07–0.81 (n = 1465; 14 studies) | 0.00–1.00 (n = 870; 13 studies) Pooled 0.51; 95% CI 0.39–0.64 Q = 120.43; $p<0.01$; $I^2 = 89\%$ | **Mild/moderate, moderate/severe, and severely ill:** Increasing prevalence with increasing severity Mild/moderate: $R^2 = 41\%$; $p = 0.004$ Moderate/Severe: $R^2 = 32\%$; $p = 0.032$ Severely ill patients: $R^2 = 18\%$; $p = 0.073$ |
| **Gastrointestinal** | 0.00–0.35 (n = 4,205; 50 studies) | 0.00 (n = 79; 2 studies) | 0.10–0.35 (n = 209; 4 studies) Pooled 0.20; 95% CI 0.09–0.32 Q = 10.96; $p = 0.01$; $I^2 = 73\%$ | 0.00–0.07 (n = 48; 3 studies) Pooled 0.07; 95% CI 0.01–0.15 Q = 1.36; $p = 0.51$; $I^2 = 0\%$ | 0.03–0.23 (n = 1181; 5 studies) Pooled 0.05; 95% CI 0.01–0.10 Q = 33.30; $p<0.01$; $I^2 = 88\%$ | 0.00–0.30 (n = 2688; 36 studies) Pooled 0.09; 95% CI 0.07–0.11 Q = 77.66; $p<0.01$; $I^2 = 55\%$ | 0.00–0.30 (n = 936; 20 studies) Pooled 0.09; 95% CI 0.06–0.11 Q = 32.34; $p = 0.03$; $I^2 = 41\%$ | 0.03–0.27 (n = 1752; 16 studies) Pooled 0.10; 95% CI 0.07–0.13 Q = 43.06; $p<0.01$; $I^2 = 65\%$ | 0.06–0.42 (n = 753; 10 studies) Pooled 0.10; 95% CI 0.03–0.19 Q = 84.40; $p<0.01$; $I^2 = 89\%$ | **Mild/moderate:** Increasing prevalence with increasing severity $R^2 = 54\%$; $p = 0.009$ **Moderate/severe and severely ill** Decreasing prevalence with increasing severity Moderate/severe: $R^2 = 23\%$; $p = 0.028$ Severely ill: $R^2 = 35\%$; $p = 0.037$ |
| **Sore throat** | 0.00–0.61 (n = 4,563; 29 studies) | 0.00 (n = 24; 1 study) | 0.05–0.40 (n = 659; 7 studies) Pooled 0.12; 95% CI 0.05–0.20 Q = 16.71; $p = 0.01$; $I^2 = 64\%$ | 0.07–0.08 (n = 28; 2 studies) Pooled 0.08; 95% CI 0.01–0.22 Q = 0.01; $p = 0.91$; $I^2 = 0\%$ | 0.04–0.46 (n = 2961; 5 studies) | 0.05–0.61 (n = 891; 14 studies) Pooled 0.14; 95% CI 0.09–0.20 Q = 51.28; $p<0.01$; $I^2 = 75\%$ | 0.08–0.26 (n = 576; 10 studies) Pooled 0.14; 95% CI 0.08–0.21 Q = 36.31; $p<0.01$; $I^2 = 75\%$ | 0.05–0.27 (n = 315; 4 studies) Pooled 0.13; 95% CI 0.05–0.25 Q = 14.87; $p<0.01$; $I^2 = 80\%$ | 0.00–0.13 (n = 226; 3 studies) Pooled 0.09; 95% CI 0.00–0.26 Q = 11.66; $p<0.01$; $I^2 = 83\%$ | – |

(Continued)

**Table 3.** (Continued)

| | Overall | Initially asymptomatic | Early | Pregnant | All-comers | Admitted to hospital | | | Severely ill patients | Source of heterogeneity identified by meta-regression or subgroup analysis in hospitalized patients |
|---|---|---|---|---|---|---|---|---|---|---|
| | | | | | | All | Mild / moderate | Moderate / severe | | |
| **Headache** | 0.00–0.53 (n = 4,335; 38 studies) | 0.00–0.07 (n = 79; 2 studies) Pooled 0.04; 95% CI 0.01–0.07 Q = 7.55; p = 0.18; I² = 34% | 0.11–0.30 (n = 217; 5 studies) Pooled 0.14; 95% CI 0.09–0.20 Q = 4.21; p = 0.38; I² = 5% | – | 0.03–0.14 (n = 2044; 4 studies) | 0.04–0.53 (n = 2105; 27 studies) Pooled 0.12; 95% CI 0.08–0.16 Q = 121.43; p<0.01; I² = 82% | 0.03–0.53 (n = 1,025; 15 studies) Pooled 0.10; 95% CI 0.06–0.16 Q = 72.87; p<0.01; I² = 81% | 0.00–0.25 (n = 1,080; 12 studies) Pooled 0.10; 95% CI 0.06–0.16 Q = 50.95; p<0.01; I² = 78% | 0.00–0.28 (n = 688; 9 studies) Pooled 0.11; 95% CI 0.07–0.16 Q = 16.32; p = 0.04; I² = 51% | – |
| **Pneumonia** | 0.11–1.00 (n = 3,086; 33 studies) | 0.50–0.67 (n = 79; 2 studies) Pooled 0.62; 95% CI 0.45–0.78 Q = 2.04; p = 0.15; I² = 51% | 0.11–0.64 (n = 66; 2 studies) | 1.00 (n = 17; 1 study) | 0.76–1.00 (n = 1675; 5 studies) | 0.54–1.00 (n = 1249; 23 studies) Pooled 0.87; 95% CI 0.82–0.92 Q = 103.36; p<0.01; I² = 79% | 0.54–1.00 (n = 869; 16 studies) Pooled 0.86; 95% CI 0.79–0.92 Q = 84.60; p<0.01; I² = 82% | 0.85–1.00 (n = 380; 7 studies) Pooled 0.91; 95% CI 0.85–0.96 Q = 12.83; p = 0.05; I² = 53% | 0.78–1.00 (n = 581; 7 studies) Out of 7 studies, 6 reported ≥0.99 | – |
| **Asymptomatic** | 0.00–0.53 (n = 46,501; 24 studies) | NA | 0.06–0.30 (n = 511; 5 studies) Pooled 0.14; 95% CI 0.11–0.17 Q = 3.35; p = 0.50; I² = 0% | 0.08–0.53 (n = 30; 2 studies) | 0.02–0.03 (n = 45,157; 4 studies) Pooled 0.02; 95% CI 0.02–0.02 Q = 2.03; p = 0.57; I² = 0% | 0.00–0.19 (n = 803; 13 studies) Pooled 0.04; 95% CI 0.01–0.07 Q = 37.23; p<0.01; I² = 68% | 0.00–0.19 (n = 593; 9 studies) Pooled 0.04; 95% CI 0.01–0.10 Q = 28.89; p<0.01; I² = 72% | 0.00–0.04 (n = 210; 4 studies) Pooled 0.01; 95% CI 0.00–0.04 Q = 3.42; p = 0.33; I² = 12% | – | – |

CI, confidence interval.

Pooled prevalence estimates and 95% CIs are shown where statistical heterogeneity was below 90% (I² <90%).

* three studies [Song F [28]; Zhou S [29]; Cheng Z [30]] with most severe mild/moderate patients have clustered with the moderate/severe studies and are analyzed as part of that group.

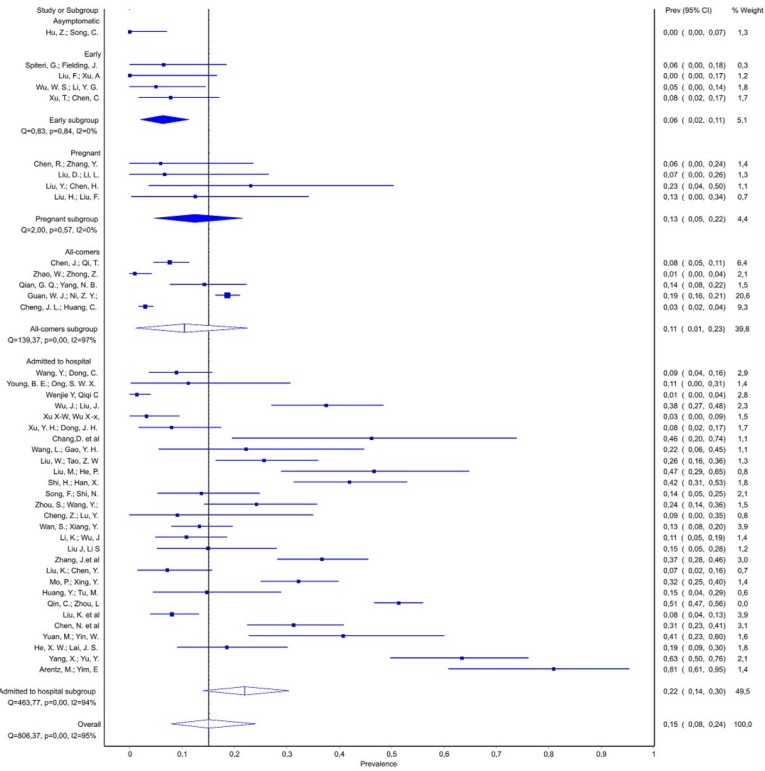

**Fig 2.**

Prevalence of asymptomatic disease significantly differed between studies in the 'early' group (14%; 95% CI 12–17%) and those in the 'all-comer' (2%; 95% CI 2–2%) or 'admitted to hospital' (4%; 95% CI 1–7%) groups (Table 3). In a study of 17 pregnant women with COVID-19 admitted to hospital for a cesarean birth, 9 were asymptomatic or had mild symptoms that did not include fever or cough [36]. In another study of 13 pregnant women admitted to hospital because of respiratory symptoms or exposure to an infected person, one patient was asymptomatic [37].

Pneumonia was common, regardless of subgroup: prevalence of at least 50% was reported in all studies except one (Table 3). This outlier, recording an 11% prevalence of pneumonia, was a study of the first recorded patients in Europe [38]. Even in two studies of initially asymptomatic patients, the pooled prevalence of pneumonia at some point during follow-up was 62% (95% 45–78%) [26, 27]. Among hospitalized patients, the pooled prevalence of pneumonia was 87% (95% CI 82–92), increasing to ≥99% in 6 out of 7 studies reporting on the severely ill subgroup. The study reporting only 78% of patients with pneumonia in critically ill patients with COVID-19 was a study of the first patients in USA [36].

## Mortality

Pooled mortality estimates were similar among studies in the 'early symptoms' (2.3%; 95% CI 0–6%), pregnant (2.6%; 95% CI 0–7%) and 'all-comer' (range 0.3% to 2.8%) subgroups (Table 3). Among studies of hospitalized patients, a distinct pattern emerged. Most studies enrolling patients with mild/moderate disease reported no deaths (7 out of 10 studies), and the remaining 3 studies reported mortality of 1–14%. However, study sample sizes and follow-up times were usually inadequate to observe those events, and the pooled estimate of 0.1% (95%

CI 0.1–4.2%) is likely biased. Studies of patients with moderate/severe disease, including sub-group of severely ill patients, reported a wide range of mortality rates, from 0–67%, with considerable heterogeneity (I2>90%, p<0.01 for both) stemming from variability in severity of patients' statuses (Table 3, Source of heterogeneity). In other words, in both moderate/severe, and severely ill patients; mortality was associated with severity of disease.

## Chest imaging

Chest imaging findings are summarized in Table 4 and forest plots are shown in the Supplementary Appendix in S1 File. The most common findings were ground glass opacity (GGO), septal thickening and consolidation. The pooled prevalence of any computed tomography (CT) finding across all studies was 89% (95% CI 83–93%), but heterogeneity was considerable. However, no study reported a prevalence of less than 50%. Even in the two studies with initially asymptomatic patients (some of whom went on to become symptomatic) the proportions of patients with any CT finding were 50% (95% CI 30–70%) and 78% (95% CI 65–89%) [26, 27]. Studies in the 'early' subgroup reported also a high prevalence of CT findings ranging from 64–88%, with the pooled proportion of 74% (95% CI 43–97%). Among patients admitted to hospital, probability of positive CT finding increased with disease severity (Table 4, Source of heterogeneity), with all the patients in the severely ill subgroup presenting with CT changes.

GGO changes were commonly reported. Apart from one study of patients with 'early' disease, which reported 8% of patients with any GGO changes [39], other studies declared that at least 40% of patients had GGO changes. In the study of 17 parturients (9 asymptomatic), all had GGO [36]. Among studies of patients admitted to hospital we found that probability of GGO was decreasing with increasing severity of illness in the mild/moderate (from ~90% to 40–47%), moderate/severe (from 98% to 67%), as well as in severely ill subgroup (from 100% to 60%) (Fig 3). However, only the associations in severely ill subgroup, and mild/moderate group reached the statistical significance (Table 3, Source of heterogeneity).

Parenchymal consolidation was less common than overall GGO changes (range 6–68%), and was lowest in a study of 108 patients with early disease [21]. Studies on patients admitted to hospital reported highly variable proportions of patients with parenchymal consolidation. The source of this heterogeneity was not evident in studies on mild/moderate patients. However, in those reporting on patients hospitalized with moderate to severe disease we identified two clusters of studies (subgroup analysis, Supplementary Appendix in S1 File). Four studies reported prevalence of consolidation between 7–19% [34, 40–42], whereas other three informed on 59–64%; with no association with the severity of disease in either cluster [29, 43, 44]. We found similar clustering pattern in the severely ill subgroup (Supplementary Appendix in S1 File). Likewise, for septal thickening, two clusters were observed among studies of patients admitted to hospital (25–37% and 63–92%), with no association with disease severity.

## Laboratory findings

Laboratory findings are summarized in Table 5 and forest plots from the quality effects model are shown in the Supplementary Appendix in S1 File. Low lymphocyte levels were reported in 20% (95% CI 11–29%) of patients in the 'initially asymptomatic' studies. In the 'early' and pregnant subgroups, pooled percentages were around 50%, although the studies were substantially heterogeneous, most likely due to differences in timing of assessment. Among studies of patients admitted to hospital, presentation with low number of lymphocytes was strongly associated with the severity of disease. Namely, in both the mild/moderate subgroup and the moderate/severe subgroup, there was a trend for increasing prevalence of lymphocytopenia with increasing severity of disease (Table 5, Source of heterogeneity). The pooled prevalence of

Table 4. Frequency of chest imaging findings, from the quality effects models, across all studies and in subgroups.

| | Overall | Initially asymptomatic | Early | Pregnant | All-comers | Admitted to hospital | | | Severely ill patients | Source of heterogeneity identified by meta-regression or subgroup analysis in hospitalized patients |
| --- | --- | --- | --- | --- | --- | --- | --- | --- | --- | --- |
| | | | | | | All | Mild / moderate | Moderate / severe | | |
| **Any CT finding** | 0.50–1.00 (n = 2,706; 40 studies) Pooled 0.89; 95% CI 0.83–0.93; Q = 359.39; $p<0.01$; $I^2$ = 89% | 0.50–0.78 (n = 74; 2 studies) Pooled 0.73; 95% CI 0.40–0.97; Q = 5.58; $p$ = 0.02; $I^2$ = 82% | 0.64–0.88 (n = 79; 2 studies) Pooled 0.74; 95% CI 0.43–0.97; Q = 5.94; $p$ = 0.01; $I^2$ = 83% | 0.81–1.00 (n = 33; 2 studies) Pooled 0.79; 95% CI 0.29–1.00; Q = 4.19; $p$ = 0.04; $I^2$ = 76% | 0.86–0.95 (n = 192; 2 studies) Pooled 0.91; 95% CI 0.82–0.98; Q = 3.78; $p$ = 0.05; $I^2$ = 73% | 0.69–1.00 (n = 2328; 32 studies) Pooled 0.90; 95% CI 0.85–0.95; Q = 257.26; $p<0.01$; $I^2$ = 88% | 0.69–1.00 (n = 1343; 20 studies) Pooled 0.89; 95% CI 0.80–0.95; Q = 181.5; $p<0.01$; $I^2$ = 89.5% | 0.85–1.00 (n = 985; 12 studies) Pooled 0.95; 95% CI 0.87–0.99; Q = 61.85; $p<0.01$; $I^2$ = 82% | 1.00–1.00 (n = X; 4 studies) Pooled 0.99; 95% CI 0.98–1.00; Q = 0.41; $p$ = 0.98; $I^2$ = 0% | **All hospitalized patients:** Increasing prevalence with increasing severity $R^2$ = 49%; $p<0.001$* |
| **Any GGO** | 0.08–1.00 (n = 1,239; 20 studies) | – | 0.08 (n = 51; 1 study) | 0.75–1.00 (n = 33; 2 studies) Pooled 0.94; 95% CI 0.61–0.99; Q = 6.02; $p$ = 0.01; $I^2$ = 83% | 0.86 (n = 101; 1 study) | 0.40–1.00 (n = 1054; 16 studies) | 0.40–0.92 (n = 617; 10 studies) | 0.67–1.00 (n = 437; 6 studies) | 0.60–1.00 (n = 186; 6 studies) Pooled 0.86; 95% CI 0.70–0.97; Q = 24.06; $p<0.01$; $I^2$ = 79% | **Mild/moderate and severely ill:** Decreasing prevalence with increasing severity Mild/moderate patients: $R^2$ = 37%; $p$ = 0.064** Severely ill patients: $R^2$ = 86%; $p$ = 0.004 |
| **Consolidation** | 0.06–0.68 (n = 1,608; 21 studies) | – | 0.06 (n = 108; 1 study) | 0.50 (n = 16; 1 study) | 0.44 (n = 101; 1 study) | 0.07–0.68 (n = 1383; 18 studies) | 0.13–0.68 (n = 800; 11 studies) | 0.07–0.64 (n = 583; 7 studies) Two clusters: Cluster 1 (0.07–0.19): Pooled 0.17; 95% CI 0.06–0.31 Q = 6.02; $p$ = 0.01; $I^2$ = 52% Cluster 2 (0.59–0.64): Pooled 0.60; 95% CI 0.54–0.65 Q = 63.99; $p$ = 0.73; $I^2$ = 0% | 0.19–0.88 (n = 132; 5 studies) Two clusters: Cluster 1 (0.19–0.24): Pooled 0.20; 95% CI 0.07–0.37 Q = 0.13; $p$ = 0.72; $I^2$ = 0% Cluster 2 (0.61–0.88): Pooled 0.68; 95% CI 0.46–0.87 Q = 7.26; $p$ = 0.72; $I^2$ = 0% | **Moderate/severe and severely ill patients:** Two clusters Moderate/severe: 0.17 (95% CI 0.06–0.31) vs. 0.60 (95% CI 0.54–0.60) Severely ill: 0.20 (95% CI 0.07–0.37) vs. 0.68 (95% CI 0.46–0.87) |

(Continued)

Table 4. (Continued)

| | Overall | Initially asymptomatic | Early | Pregnant | All-comers | Admitted to hospital | | | Severely ill patients | Source of heterogeneity identified by meta-regression or subgroup analysis in hospitalized patients |
| | | | | | | All | Mild / moderate | Moderate / severe | | |
|---|---|---|---|---|---|---|---|---|---|---|---|
| **Septal thickening** | 0.25–0.92 (n = 739; 8 studies) | – | – | – | – | 0.25–0.92 (n = 739; 8 studies) Two clusters Cluster 1 (0.25–0.37): Pooled 0.35; 95% CI 0.30–0.40 Q = 3.35; p = 0.34; $I^2$ = 10% Cluster 2 (0.63–0.92): Pooled 0.75; 95% CI 0.63–0.86 Q = 7.31; p = 0.06; $I^2$ = 59% | 0.35–0.92 (n = 583; 6 studies) | 0.25–0.63 (n = 156; 2 studies) | 0.43–0.92 (n = 63;3 studies): 0.43 (n = 14; 1 all-comers study) 0.76–0.92 (n = 49; 2 moderate/severe studies) | **All hospitalized patients:** Two clusters 0.35 (95% CI 0.30–0.40) vs. 0.75 (95% CI 0.63–0.86) |

CI, confidence interval; CT, computed tomography; GGO, ground glass opacity.

Pooled prevalence estimates and 95% CIs are shown where there was appropriately low heterogeneity ($I^2$<90).

* statistically significant at 0.1 level,

**heteroscedastic maximum likelihood based estimation procedure.

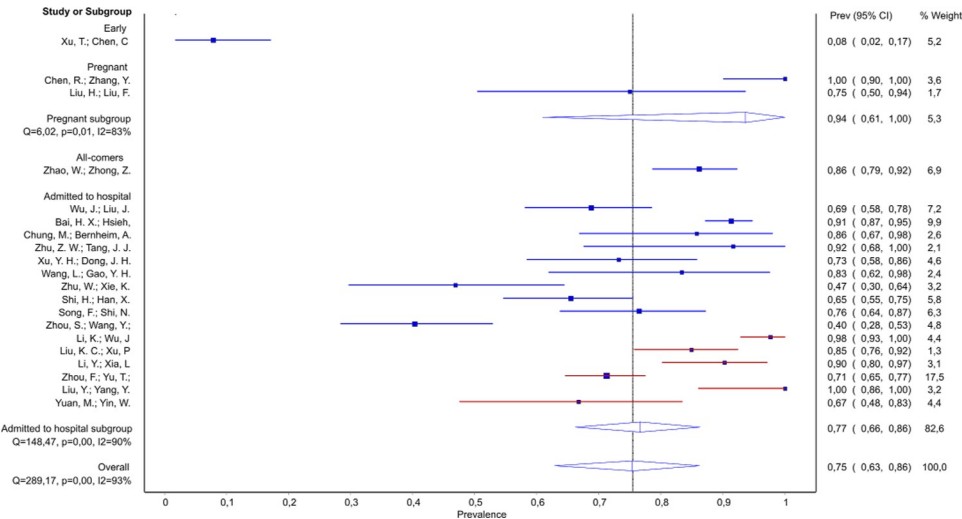

**Fig 3.**

lymphocytopenia among severely ill patients was high and reached 84% (95% CI 77–89%). Among severely ill patients with lymphocytopenia, the pooled estimate of lymphocyte level was 0.62 (95% CI 0.58–0.66) x10$^9$ cells/L (heterogeneity, Q = 1.8; $p$ = 0.41; I$^2$ = 0), but among severely ill patients overall there was considerable heterogeneity in lymphocyte count (0.67–0.90; Q = 112.56; $p$ = 0.00; I$^2$ = 96%).

Low white blood cell (WBC) count was described in approximately 20% of patients in the 'initially asymptomatic' studies and the 'early' studies. Among studies including hospitalized patients, there was a trend for a decrease in prevalence of low WBC count along with increasing severity of disease among all hospitalized (from ~30% to ~10%), as well as moderate/severe hospitalized patients (from ~20% to ~10%) (Fig 4A and Table 5, Source of heterogeneity). While the initial meta-regression analysis across all studies on hospitalized patients did not reach significance, sensitivity analysis identified the study by Bai et al. as a source of heterogeneity [49]. After removal of this study, we found a significant trend across the remaining 20 studies. Complementary data were found for elevated WBC count for which we also found association with the disease severity (Fig 4B and Table 5, Source of heterogeneity), Notably, we observed an increase in prevalence with increasing severity of the disease among moderate/severe hospitalized patients, and also found that probability of elevated WBC was more likely in moderate/severe patients than in mild moderate: 0.17 (95% CI 0.12–0.23) vs. 0.03 (95% CI 0.02–0.05). Studies in the mild/moderate subgroup initially showed considerable heterogeneity. However, all of this heterogeneity was derived from the study by Bai et al., which reported 29% of patients with elevated WBC [49]. Exclusion of this study mitigated heterogeneity and a pooled prevalence of 3% (95% CI 1.8–4.9) was estimated for this subgroup. The results from low and elevated WBC analyses suggest that these proportions are interchanged in the Bai et al. paper [42].

Elevated C-reactive protein (CRP) was one of the most common laboratory findings. The proportion of patients with elevated CRP in the 'initially asymptomatic' studies (pooled prevalence 19%; 95% CI 10–28%) was significantly lower than studies in hospitalized patients (63%, 95% CI 0.48–0.77). Moreover, in patients admitted to hospital, elevated CRP was further associated with severity of disease as severely ill patients presented with elevated CRP more often than mild/moderate group: 0.88 (95% CI 0.76–0.98) vs. 0.63 (95% CI 0.48–0.77). Finally, in the

**Table 5. Frequency of laboratory findings, from the quality effects models, across all studies and in study subgroups.**

| | Overall | Initially asymptomatic | Early | Pregnant | All-comers | Admitted to hospital | | | Severely ill patients | Source of heterogeneity identified by meta-regression |
|---|---|---|---|---|---|---|---|---|---|---|
| | | | | | | All | Mild / moderate | Moderate / severe | | |
| **ALT—elevated** | 0.00–0.41 (n = 1,966; 16 studies) Pooled 0.22; 95% CI 0.14–0.32 Q = 134.39; $I^2$ = 89%; $p<0.01$ | 0.00–0.08 (n = 79; 2 studies) Pooled 0.02; 95% CI 0.00–0.13 Q = 4.36; $p = 0.04$; $I^2$ = 77% | – | – | 0.21 (n = 741; 1 study) | 0.04–0.41 (n = 1146; 13 studies) Pooled 0.26; 95% CI 0.18–0.35 Q = 84.87; $I^2$ = 86%; $p<0.01$ | 0.04–0.28 (n = 263; 4 studies) Pooled 0.11; 95% CI 0.03–0.22 Q = 12.22; $p = 0.01$; $I^2$ = 75% | 0.17–0.41 (n = 883; 9 studies) Pooled 0.31; 95% CI 0.25–0.38 Q = 23.16; $I^2$ = 65%; $p<0.01$ | 0.28–0.41 (n = 388; 3 studies): (n = 135; 1 all-comers study) 0.28 (n = 49; 2 moderate/severe studies) Pooled 0.40; 95% CI 0.34–0.46 Q = 0.01; $p = 0.93$; $I^2$ = 0% | **All hospitalized patients:** More moderate/severe patients present with elevated ALT than mild/moderate 0.31 (95% CI 0.25–0.38) vs. 0.11 (95% CI 0.02–0.22) Increasing prevalence with increasing severity $R^2$ = 52%; $p = 0.006$ |
| **AST—elevated** | 0.00–0.61 (n = 2,026; 17 studies) Pooled 0.26; 95% CI 0.17–0.36 Q = 97.70; $I^2$ = 87%; $p<0.01$ | 0.00 (n = 24; 1 study) | – | – | 0.10–0.22 (n = 848; 2 studies) Pooled 0.21; 95% CI 0.06–0.39 Q = 8.78; $I^2$ = 89%; $p<0.01$ | 0.16–0.61 (n = 1154; 14 studies) Pooled 0.26; 95% CI 0.17–0.36 Q = 44.69; $I^2$ = 78%; $p<0.01$ | 0.16–0.25 (n = 325; 5 studies) Pooled 0.14; 95% CI 0.06–0.25 Q = 16.25; $I^2$ = 75%; $p<0.01$ | 0.17–0.61 (n = 829; 9 studies) Pooled 0.33; 95% CI 0.25–0.42 Q = 36.24; $I^2$ = 78%; $p<0.01$ | 0.33–0.62 (n = 454; 6 studies) Pooled 0.42; 95% CI 0.34–0.51 Q = 10.74; $p = 0.06$; $I^2$ = 53% | **All hospitalized patients:** More moderate/severe patients present with elevated AST than mild/moderate 0.33 (95% CI 0.25–0.42) vs. 0.14 (95% CI 0.06–0.25) Increasing prevalence with increasing severity $R^2$ = 59%; $p = 0.001$ |
| **CRP—elevated** | 0.17–1.00 (n = 2,305; 26 studies) | 0.17–0.18 (n = 78; 2 studies) Pooled 0.19; 95% CI 0.10–0.28 Q = 0.0; $I^2$ = 0%; $p>0.99$ | 0.30–0.99 (n = 118; 2 studies) | 0.41–0.81 (n = 48; 3 studies) Pooled 0.59; 95% CI 0.34–0.82 Q = 5.52; $p = 0.06$; $I^2$ = 64% | 0.54–0.61 (n = 884; 2 studies) Pooled 0.60; 95% CI 0.54–0.66 Q = 1.57; $I^2$ = 36%; $p = 0.21$ | 0.38–1.00 (n = 1177; 17 studies) | 0.38–1.00 (n = 559; 10 studies) Pooled 0.63; 95% CI 0.48–0.77 Q = 93.14; $I^2$ = 89.9%; $p<0.01$ | 0.60–0.92 (n = 618; 7 studies) Pooled 0.80; 95% CI 0.68–0.91 Q = 58.74; $I^2$ = 89.8%; $p<0.01$ | 0.40–1.00 (n = 320; 7 studies) Pooled 0.88; 95% CI 0.76–0.98 Q = 26.26; $p<0.01$; $I^2$ = 77% | **All hospitalized patients:** More severely ill patients present with elevated CRP than mild/moderate 0.88 (95% CI 0.76–0.98) vs. 0.63 (95% CI 0.48–0.77) **Moderate/severe patients:** Increasing prevalence with increasing severity $R^2$ = 95%; $p<0.001$ (excluding Zhang J et al. 2020 [45] based on sensitivity analysis) |

*(Continued)*

Table 5. (Continued)

| | Overall | Initially asymptomatic | Early | Pregnant | All-comers | Admitted to hospital | | | Severely ill patients | Source of heterogeneity identified by meta-regression |
|---|---|---|---|---|---|---|---|---|---|---|
| | | | | | | All | Mild / moderate | Moderate / severe | | |
| **D-dimer— elevated** | 0.04–0.46 (n = 1,657; 12 studies) | 0.18 (n = 22; 1 study) | – | – | 0.08–0.46 (n = 651; 2 studies) | 0.04–0.43 (n = 985; 9 studies) | 0.04–0.14 (n = 261; 3 studies) Pooled 0.09; 95% CI 0.03–0.18 Q = 6.74; p = 0.03; I² = 70% | 0.14–0.43 (n = 724; 6 studies) Pooled 0.29; 95% CI 0.18–0.42 Q = 48.60; p<0.01; I² = 89.7% | 0.14–0.61 (n = 296; 3 studies) (n = 147; 2 studies) Pooled 0.60; 95% CI 0.52–0.68 Q = 0.0006; p = 0.94; I² = 0% 0.14 (n = 149; one study with the most severely ill patients that was excluded from meta-regression of all hospitalized studies) | **All hospitalized patients:** More moderate/severe patients present with elevated D-dimer than mild/moderate 0.29 (95% CI 0.18–0.42) vs. 0.09 (95% CI 0.03–0.18) Increasing prevalence with increasing severity R² = 67%; p = 0.009 (excluding two studies with the most severe patients Wu, C [46] and Yang, X. [47]) |
| **ESR— elevated** | 0.09–0.92 (n = 647; 9 studies) | 0.36 (n = 55; 1 study) | – | – | – | 0.09–0.92 (n = 592; 8 studies) | 0.09–0.74 (n = 219; 4 studies) | 0.59–0.92 (n = 373; 4 studies) Pooled 0.87; 95% CI 0.75–0.95 Q = 16.44; p<0.01; I² = 82% | 0.81 (n = 199; 1 study) | **All hospitalized patients:** Increasing prevalence with increasing severity R² = 49%; p = 0.079 (excluding Zhao, X. et al. 2020 [48]) |
| **IL-6— elevated** | 0.00–0.94 (n = 480; 6 studies) | 0.00 (n = 55; 1 study) | – | – | – | 0.27–0.94 (n = 425; 5 studies) | 0.27 (n = 26; 1 study) | 0.38–0.94 (n = 399; 4 studies) | 0.76–0.94 (n = 75; 2 studies) Pooled 0.91; 95% CI 0.69–1.00 Q = 4.46; p = 0.03; I² = 78% | **All hospitalized patients:** Increasing prevalence with increasing severity R² = 70%; p = 0.086* |
| **Lymphocytes —low** | 0.17–0.89 (n = 2,926; 30 studies) | 0.17–0.20 (n = 79; 2 studies) Pooled 0.20; 95% CI 0.11–0.29 Q = 0.08; p = 0.79; I² = 0% | 0.20–0.60 (n = 118; 2 studies) Pooled 0.50; 95% CI 0.09–0.91 Q = 5.89; p = 0.02; I² = 83% | 0.29–0.80 (n = 48; 3 studies) Pooled 0.54; 95% CI 0.24–0.84 Q = 8.24; p = 0.02; I² = 76% | 0.83 (n = 879; 1 study) | 0.28–0.89 (n = 1802; 22 studies) | 0.28–0.84 (n = 753; 11 studies) | 0.35–0.89 (n = 1049; 11 studies) | 0.80–0.89 (n = 175; 4 studies) Pooled 0.84; 95% CI 0.77–0.89 Q = 1.9; p = 0.60; I² = 0% | **Mild/moderate and Moderate/severe:** Increasing prevalence with increasing severity Mild/moderate more severe patients: R² = 78; p = 0.026 (6 studies) Moderate/severe more severe patients: R² = 67%; p = 0.030 (7 studies) |

(Continued)

Table 5. (Continued)

| | Overall | Initially asymptomatic | Early | Pregnant | All-comers | Admitted to hospital | | | Severely ill patients | Source of heterogeneity identified by meta-regression |
| --- | --- | --- | --- | --- | --- | --- | --- | --- | --- | --- |
| | | | | | | All | Mild / moderate | Moderate / severe | | |
| Lymphocytes—elevated | 0.00–0.50 (n = 506; 6 studies) | – | 0.40 (n = 108; 1 study) | 0.00 (n = 15; 1 study) | – | 0.00–0.50 (n = 383; 4 studies) | – | 0.00–0.50 (n = 383; 4 studies) | – | **Moderate/severe:** Decreasing prevalence with increasing severity $R^2$ = 81%; $p$ = 0.098* |
| Neutrophils—low | 0.00–0.23 (n = 636; 8 studies) Pooled 0.06; 95% CI 0.01–0.29 Q = 22.77; $p<0.01$; $I^2$ = 69% | 0.20 (n = 55; 1 study) | – | – | 0.11 (n = 91; 1 study) | 0.00–0.23 (n = 490; 6 studies) | 0.00–0.23 (n = 229; 2 studies) | 0.00–0.03 (n = 261; 4 studies) Pooled 0.02; 95% CI 0.00–0.05 Q = 8.33; $p = 0.04$; $I^2$ = 64% | 0.08 (n = 25; 1 study) | – |
| Neutrophils—elevated | 0.03–0.61 (n = 811; 11 studies) | – | – | – | 0.03 (n = 91; 1 study) | 0.04–0.61 (n = 720; 10 studies) | 0.04–0.61 (n = 295; 5 studies) | 0.12–0.38 (n = 425; 5 studies) Pooled 0.30; 95% CI 0.20–0.42 Q = 15.93; $p<0.01$; $I^2$ = 75% | 0.28 (n = 83; 1 study) | – |
| PCT—elevated | 0.00–0.53 (n = 947; 13 studies) | 0.21(n = 24; 1 study) | – | – | 0.15 (n = 91; 1 study) | 0.00–0.53 (n = 832; 11 studies) | 0.00–0.25 (n = 190; 4 studies) Pooled 0.05; 95% CI 0.00–0.15 Q = 15.63; $p<0.01$; $I^2$ = 81% | 0.01–0.53 (n = 642; 7 studies) | 0.10–0.50 (n = 226; 5 studies) Pooled 0.22; 95% CI 0.07–0.40 Q = 26.62.; $p<0.01$; $I^2$ = 85% | – |
| Platelets—low | 0.05–0.36 (n = 1,829; 12 studies) | – | – | – | 0.11–0.36 (n = 960; 2 studies) | 0.05–0.26 (n = 869; 10 studies) Pooled 0.12; 95% CI 0.09–0.16 Q = 20.86; $p = 0.01$; $I^2$ = 57% | 0.05–0.14 (n = 291; 3 studies) Pooled 0.12; 95% CI 0.07–0.18; Q = 4.09; $p = 0.13$; $I^2$ = 51% | 0.07–0.26 (n = 578; 7 studies) Pooled 0.12; 95% CI 0.07–0.18; Q = 16.49; $p = 0.01$; $I^2$ = 64% | – | **Moderate/severe patients:** Decreasing prevalence with increasing severity $R^2$ = 61%; $p$ = 0.016 |
| Platelets—elevated | 0.00–0.05 (n = 498; 6 studies) | – | – | – | 0.03 (n = 91; 1 study) | 0.00–0.05 (n = 407; 5 studies) Pooled 0.02; 95% CI 0.00–0.06 Q = 11.13; $p = 0.03$; $I^2$ = 64% | 0.00–0.05 (n = 229; 2 studies) | 0.00–0.04 (n = 178; 3 studies) Pooled 0.03; 95% CI 0.00–0.07 Q = 3.75; $p = 0.15$; $I^2$ = 47% | – | – |

*(Continued)*

**Table 5.** (Continued)

| | Overall | Initially asymptomatic | Early | Pregnant | All-comers | Admitted to hospital | | | Severely ill patients | Source of heterogeneity identified by meta-regression |
|---|---|---|---|---|---|---|---|---|---|---|
| | | | | | | All | Mild / moderate | Moderate / severe | | |
| **Troponin** | 0.07–0.17 (n = 181; 3 studies) | – | – | – | – | Only data on moderate/severe patients were available | – | 0.07–0.17 (n = 181; 3 studies) Pooled 0.16; 95% CI 0.11–0.22 Q = 0.74; p = 0.69; I² = 0% | – | – |
| **WBC—low** | 0.00–0.51 (n = 3,364; 29 studies) | 0.17–0.20 (n = 79; 2 studies) Pooled 0.19; 95% CI 0.11–0.29 Q = 0.07; p = 0.79; I² = 0% | 0.10–0.33 (n = 169; 3 studies) Pooled 0.18; 95% CI 0.03–0.39 Q = 11.80; p<0.01; I² = 83% | – | 0.15–0.34 (n = 1204; 3 studies) Pooled 0.31; 95% CI 0.16–0.48 Q = 18.29; p = <0.01; I² = 89% | 0.00–0.51 (n = 1912; 21 studies) | 0.00–0.51 (n = 777; 10 studies) | 0.00–0.37 (n = 1135; 11 studies) Pooled 0.18; 95% CI 0.12–0.24 Q = 55.96; p<0.01; I² = 82% | 0.08–0.16 (n = 308; 4 studies) Pooled 0.11; 95% CI 0.08–0.15 Q = 1.6; p = 0.659; I² = 0% | **All hospitalized and moderate/severe patients:** Decreasing prevalence with increasing severity All hospitalized: R² = 32%; p = 0.012 (excluding Bai H et al. 2020 [49]) Moderate/severe: R² = 18%; p = 0.043 |
| **WBC—elevated** | 0.00–0.50 (n = 3,341; 26 studies) | 0.02 (n = 55; 1 study) | 0.00 (n = 108; 1 study) | 0.50 (n = 16; 1 study) | 0.03–0.06 (n = 1069; 2 studies) Pooled 0.06; 95% CI 0.04–0.07 Q = 0.88; p = 0.35; I² = 0% | 0.00–0.35 (n = 2093; 21 studies) | 0.00–0.29 (n = 747; 9 studies) Pooled 0.03; 95% CI 0.02–0.05 Q = 6.08; p = 0.53; I² = 0% (excluding Bai H et al. 2020 [49]) | 0.03–0.35 (n = 1346; 12 studies) Pooled 0.17; 95% CI 0.12–0.23 Q = 55.03; p<0.01; I² = 80% | 0.08–0.54 (n = 362; 5 studies) Pooled 0.21; 95% CI 0.10–0.35 Q = 17.87; p<0.01; I² = 78% | **All hospitalized patients:** More moderate/severe patients present with elevated WBC than mild/moderate 0.17 (95% CI 0.12–0.23) vs. 0.03 (95% CI 0.02–0.05) **Moderate/severe:** Increasing prevalence with increasing severity R² = 61%; p = 0.002 (excluding Bai H et al. 2020 [49]). |

ALT, alanine aminotransferase; AST, aspartate aminotransferase; CI, confidence interval; CRP, C-reactive protein; PCT, procalcitonin; IL-6, interleukin-6; WBC, white blood cell count.

Pooled prevalence estimates and 95% CIs are shown where there was appropriately low heterogeneity (I²<90%).

* significance at 0.1 level.

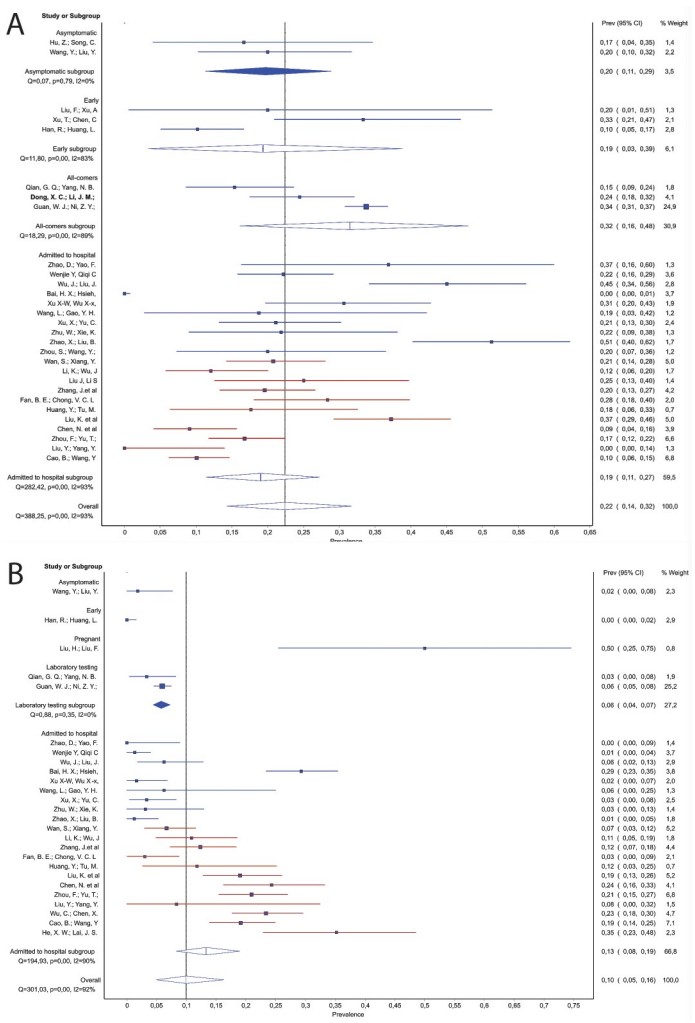

**Fig 4.**

moderate/severe group of studies we observed an increase in the prevalence of elevated CRP as disease severity increased, from ~60% to 91% (Table 5, Source of heterogeneity). For interleukin (IL)-6, among patients admitted to hospital, we were able to show that the proportion of patients with interleukin (IL)-6 values above 7 pg/ml appears to increase with the severity of illness, although this only included 5 studies (Fig 5 and Table 5, Source of heterogeneity). The single 'initially asymptomatic' subgroup study that reported this variable did not find elevated IL-6 values in any patients (n = 55) [27]. Among hospitalized patients, elevated D-dimer levels were detected more frequently in the moderate/severe than the mild/moderate group: 0.29 (95% CI 0.18–0.42) vs 0.09 (95% CI 0.03–0.18). In severely ill subgroup, this parameter reached the pooled proportion of 0.60, 95% CI 0.52–0.68 after exclusion of severely ill patients from an outlier 'all-comers' study. Low platelet levels were recorded in 12% of hospitalized patients (95% CI 9–35%). Yet, in moderate/severe group we observed a decreasing prevalence of patients with low platelet count, from ~20% to ~5%, with the increasing severity of COVID19. Elevated troponin was recorded in 16% (95% CI 11–22) of patients in moderate/severe group, but no data were published on other groups.

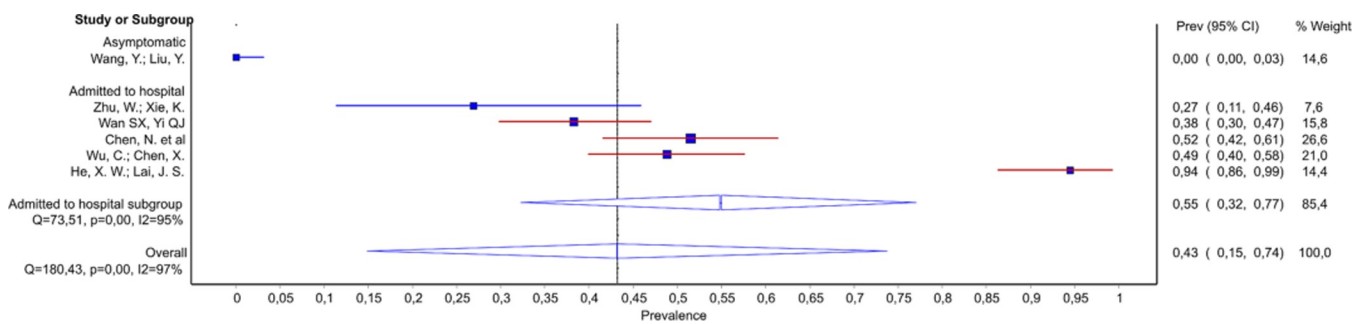

**Fig 5.**

Alanine aminotransferase (ALT) was rarely elevated among 'initially asymptomatic' patients (pooled prevalence of 2%; 95% CI 0–13%), and in a single 'all-comer' study the prevalence was 21% (95% CI 18–24%). Among studies of hospitalized patients, prevalence of patients with elevated ALT increased with increasing severity of disease, from 4–41% (Table 5, Source of heterogeneity). Moderate/severe group presented with more elevated ALT than mild/moderate 0.31 (95% CI 0.25–0.38) vs. 0.11 (95% CI 0.02–0.22); while the pooled proportion among severely ill patients without the all-comers study [47] was 40% (95% CI 34–46%). Similarly, across all subgroups there was a clear increase in the occurrence of elevated aspartate aminotransferase (AST) as disease severity increased (Fig 6), with a significant trend detected across the studies of hospitalized patients, from 16–61% ($R^2$ = 59%; $p<0.001$).

## Hospital admissions and supportive therapy

Duration of hospitalization was reported in 8 studies (n = 1,653) and ranged from 3.7 to 23.2 days. Use of supportive therapy is summarized in Table 6. Overall up to 30% of patients were admitted to an ICU, and the pooled proportion of ICU admissions among the all-comer studies (n = 1439) was 6% (95% 3–10%). Invasive ventilation was used in 0–71% of patients overall. In the subgroup of severely ill patients 3–71% had invasive ventilation and there was a strong trend for increasing use with increased severity (Table 6, Source of heterogeneity). Non-

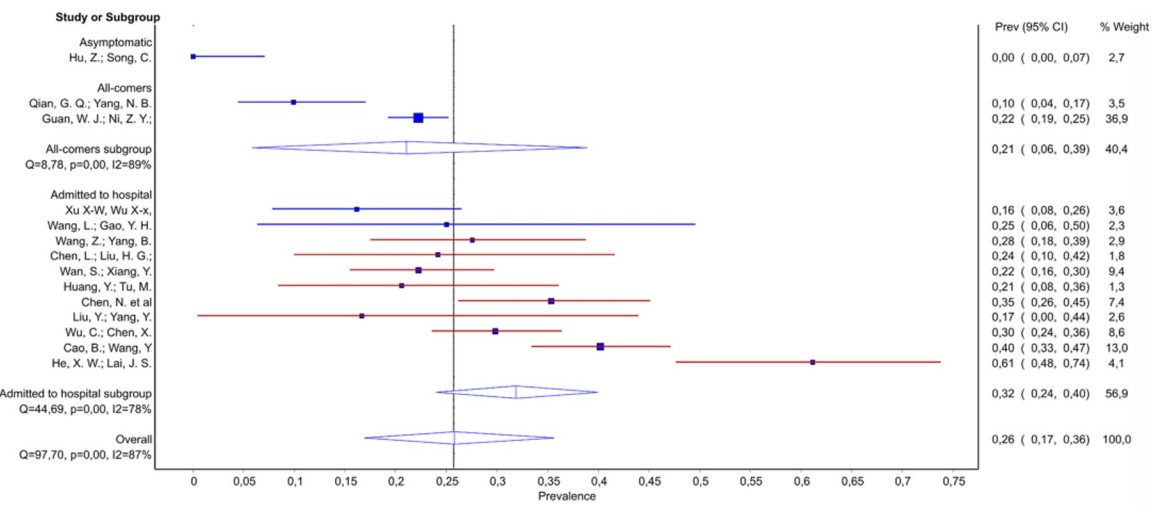

**Fig 6.**

**Table 6. Frequency of treatment and supportive therapy, from the quality effects models, overall and in subgroups of studies.**

| | Overall | Initially asymptomatic | Early | Pregnant | All-comers | Admitted to hospital | | | Severely ill patients | Source of heterogeneity identified by meta-regression |
| --- | --- | --- | --- | --- | --- | --- | --- | --- | --- | --- |
| | | | | | | All | Mild / moderate | Moderate / severe | | |
| **Antibiotics** | 0.04–1.00 (n = 2471; 17 studies) | 0.04 (n = 24; 1 study) | 0.20 (n = 10; 1 study) | 1.00 (n = 15; 1 study) | 0.58 (n = 1099; 1 study) | 0.23–1.00 (n = 1323; 13 studies) | 0.23–1.00 (n = 321; 4 studies) | 0.44–0.98 (n = 1002; 9 studies) | 0.20–1.00 (n = 676; 9 studies) Pooled 0.90; 95% CI 0.80–0.97 Q = 62.85; $p<0.01$; $I^2$ = 87% | **All hospitalized patients:** Increasing probability of antibiotics given with increasing severity $R^2$ = 49%; $p$ = 0.018 |
| **Antivirals** | 0.21–1.00 (n = 3059; 27 studies) | 0.88–1.00 (n = 79; 2 studies) | 1.00 (n = 10; 1 study) | 0.73 (n = 15; 1 study) | 0.36 (n = 1099; 1 study) | 0.21–1.00 (n = 1856; 22 studies) | 0.28–1.00 (n = 358; 6 studies) | 0.21–1.00 (n = 1498; 16 studies) | 0.44–1.00 (n = 725; 12 studies) | **All hospitalized patients:** Decreasing probability of antivirals given with increasing severity $R^2$ = 19%; $p$ = 0.044 |
| **Immunoglobulin** | 0.04–0.73 (n = 2499; 18 studies) | 0.04–0.13 (n = 79; 2 studies) | 0.50 (n = 10; 1 study) | – | 0.13 (n = 1099; 1 study) | 0.06–0.73 (n = 1311; 14 studies) | 0.13–0.33 (n = 321; 4 studies) Pooled 0.19; 95% CI 0.11–0.28 Q = 9.5; $p$ = 0.02; $I^2$ = 68% | 0.06–0.73 (n = 990; 10 studies) | 0.16–0.54 (n = 335; 4 studies) Pooled 0.36; 95% CI 0.22–0.51 Q = 18.26; $p<0.01$; $I^2$ = 84% | **All hospitalized patients:** Increasing probability of immunoglobulin given with increasing severity $R^2$ = 20%; $p$ = 0.034 |
| **Steroids** | 0.00–0.83 (n = 2917; 22 studies) | 0.00–0.04 (n = 79; 2 studies) | 0.30 (n = 10; 1 study) | – | 0.19 (n = 1099; 1 study) | 0.00–0.83 (n = 1729; 18 studies) | 0.00–0.51 (n = 387; 5 studies) | 0.19–0.83 (n = 1342; 13 studies) | 0.34–0.83 (n = 655; 8 studies) Pooled 0.52; 95% CI 0.39–0.65 Q = 51.83; $p<0.01$; $I^2$ = 86% | **All hospitalized patients:** Increasing probability of steroids given with increasing severity $R^2$ = 36%; $p$ = 0.004 |
| **ICU admission** | 0.00–0.30 (n = 2410; 17 studies) | 0.00–0.00 (n = 79; 2 studies) | 0.30 (n = 10; 1 study) | 0.08 (n = 13; 1 study) | 0.05–0.10 (n = 1439; 3 studies) Pooled 0.06; 95% CI 0.03–0.10 Q = 7.50; $p$ = 0.02; $I^2$ = 73% | 0.00–0.26 (n = 869; 10 studies) | 0.00–0.11 (n = 277; 5 studies) Pooled 0.02; 95% CI 0.00–0.06 Q = 10.51; $p$ = 0.03; $I^2$ = 62% | 0.13–0.26 (n = 592; 5 studies) Pooled 0.24; 95% CI 0.20–0.29 Q = 5.69; $p$ = 0.22; $I^2$ = 30% | 0.19–1.00 (n = 99; 3 studies) | **All hospitalized patients:** ICU admission is 22% more likely in moderate/severe than in mild moderate patients 0.24 (95% CI 0.20–0.29) vs. 0.02 (0–0.06) Increasing occurrence with increasing severity $R^2$ = 77%; $p$ = 0.003 |
| **Oxygen** | 0.00–0.96 (n = 2928; 21 studies) | 0.04 (n = 55, 1 study) | 0.90 (n = 10; 1 study) | 0.93 (n = 15; 1 study) | 0.41 (n = 1099; 1 study) | 0.00–0.96 (n = 1749; 17 studies) | 0.00–0.94 (n = 537; 5 studies) | 0.05–0.96 (n = 1212; 17 studies) | – | **All hospitalized patients:** Two non-overlapping clusters are identified (0.05–0.33) and (0.49–0.96) both showing decrease in prevalence with increasing severity Cluster 1 did not reach significance Cluster 2 $R^2$ = 42%; $p$ = 0.033 |

*(Continued)*

**Table 6.** (Continued)

| | Overall | Initially asymptomatic | Early | Pregnant | All-comers | Admitted to hospital | | | Severely ill patients | Source of heterogeneity identified by meta-regression |
| --- | --- | --- | --- | --- | --- | --- | --- | --- | --- | --- |
| | | | | | | All | Mild / moderate | Moderate / severe | | |
| **Non-invasive ventilation** | 0.00–0.56 (n = 2788; 24 studies) | 0.00 (n = 24; 1 study) | 0.00 (n = 10; 1 study) | 0.00 (n = 13; 1 study) | 0.05 (n = 1099; 1 study) | 0.00–0.56 (n = 1642; 20 studies) | 0.00–0.44 (n = 355; 5 studies) | 0.05–0.56 (n = 1287; 15 studies) Pooled 0.23; 95% CI 0.16–0.32 Q = 112.78; $p<0.01$; $I^2$ = 88% | 0.05–0.73 (n = 732; 10 studies) | **All hospitalized patients:** Increasing prevalence with increasing severity $R^2$ = 22%; $p = 0.038$ |
| **Invasive ventilation** | 0.00–0.71 (n = 2907; 25 studies) | 0.00–0.00 (n = 79; 2 studies) Pooled 0.01; 95% CI 0.0–0.03 Q = 0.08; $p = 0.78$; $I^2$ = 0% | 0.00–0.00 (n = 61; 2 studies) Pooled 0.01; 95% CI 0.0–0.04 Q = 0.24; $p = 0.62$; $I^2$ = 0% | 0.08 (n = 13; 1 study) | 0.02 (n = 1099; 1 study) | 0.00–0.71 (n = 1655; 19 studies) | 0.00–0.02 (n = 417; 6 studies) | 0.00–0.71 (n = 1238; 13 studies) | 0.03–0.71 (n = 692; 9 studies) | **All hospitalized patients:** Increasing prevalence with increasing severity $R^2$ = 48%; $p = 0.003$ **Severely ill patients:** Increasing prevalence with increasing severity $R^2$ = 74%; $p = 0.004$ |
| **ECMO** | 0.00–0.12 (n = X; 15 studies) Pooled 0.011; 95% CI 0.002–0.025 Q = 36.23; $p<0.01$; $I^2$ = 61% | 0.00–0.00 (n = 79; 2 studies) Pooled 0.006; 95% CI 0.00–0.029 Q = 0.08; $p = 0.78$; $I^2$ = 0% | – | 0.08; 95% CI 0.00–0.30 (n = 13; 1 study) | 0.00 (n = 1099; 1 study) | 0.00–0.12 (n = 1154; 11 studies) Pooled 0.016; 95% CI 0.005–0.033 Q = 26.60; $p<0.01$; $I^2$ = 62% | 0.00–0.00 (n = X; 3 studies) Pooled 0.004; 95% CI 0.00–0.017 Q = 4.28; $p = 0.23$; $I^2$ = 30% | 0.00–0.12 (n = X; 7 studies) Pooled 0.024; 95% CI 0.008–0.046 Q = 16.85; $p = 0.02$; $I^2$ = 58% | 0.00–0.17 (n = X; 9 studies) Pooled 0.03; 95% CI 0.01–0.06 Q = 15.33; $p = 0.05$; $I^2$ = 48% | **All hospitalized patients:** Increasing usage of ECMO with increasing severity $R^2$ = 41%; $p = 0.013$ |
| **Dialysis** | 0.00–0.09 (n = 534; 7 studies) | 0.00 (n = 24; 1 study) | – | – | – | 0.00–0.09 (n = 510; 6 studies) Pooled 0.054; 95% CI 0.025–0.093 Q = 10.28; $p = 0.07$; $I^2$ = 51% | 0.00–0.01 (n = 110; 2 studies) Pooled 0.016; 95% CI 0.0–0.043 Q = 0.17; $p = 0.68$; $I^2$ = 0% | 0.05–0.09 (n = 400; 4 studies) Pooled 0.07; 95% CI 0.046–0.098 Q = 2.16; $p = 0.54$; $I^2$ = 0% | 0.05–0.10 (n = X; 4 studies) Pooled 0.08; 95% CI 0.05–0.12 Q = 2.10; $p = 0.55$; $I^2$ = 0% | **All hospitalized patients:** Dialysis is 5% more likely in moderate/severe than in mild moderate patients 0.07 (95% CI 0.046–0.098) vs. 0.016 (0.0–0.043) |

ECMO, extracorporeal membrane oxygenation.

Pooled prevalence estimates and 95% CIs are shown where there was appropriately low heterogeneity ($I^2<90$).

invasive ventilation was used in 0–56% of patients (5–71% in severely ill patients) and dialysis in 0–9% (8%; 95% CI 5–12% in severely ill). Extracorporeal membrane oxygenation (ECMO) was used in 1% (95% CI 0–2%) of patients overall and in 3% of severely ill patients (95% CI 1–6%).

## Pharmacological treatment approaches

Most studies were descriptive, with no control group, and therefore did not assess the efficacy and safety of interventions. Use of any antiviral was specifically mentioned in 27 studies (n = 3,059), with most of the studies reporting at least 73% of patients receiving at least one antiviral (oseltamivir, ritonavir, lopinavir, ribavirin, peramivir and umifenovir). Six studies reported that 21–50% of patients received any antivirals [24, 25, 47, 50–52]. Only one RCT was identified, an open-label trial in 199 Chinese COVID-19 patients in which 99 patients were assigned to the lopinavir–ritonavir group (400 mg and 100 mg orally, for 14 days) plus standard care, and 100 patients to the standard care group [24]. Standard care comprised, as necessary, supplemental oxygen, noninvasive and invasive ventilation, antibiotic agents, vasopressor support, renal-replacement therapy, and ECMO. There were no differences between the two groups in terms of time to clinical improvement (hazard ratio for clinical improvement, 1.31; 95% CI 0.95–1.80), mortality at 28 days (19.2% vs 25.0%; difference, −5.8 percentage points; 95% CI −17.3–5.7) or proportion of patients with detectable viral RNA. Gastro-intestinal adverse events were more common in the intervention group, but serious adverse events were more common in the control group (64.6% vs. 37.9%, mostly respiratory failure, acute kidney damage, and secondary infection). Among 17 studies that reported antibiotic use (n = 2,471), 4–95% of patients received this treatment (including beta-lactams, fluoroquinolones, cephalosporins, piperacillin-tazobactam and meropenem). The probability of treatment increased with severity of illness and among hospitalized patients there was an increasing trend, from 23–100% (Table 6, Source of heterogeneity), with 90% (95% CI 80–97) of severely ill patients receiving antibiotics. Taking steroids also depended on severity of illness (Table 6, Source of heterogeneity) with 0–3% steroids received by least severe hospital patients, followed by 30% that was reported in majority of moderately/severe studies, and finally 58–65% taken by most severely ill patients (22 studies; n = 2917). Four studies reported steroid use that was much higher than anticipated based on the severity of included patients [35, 51, 53, 54]. Immunoglobulin usage (18 studies; n = 2,499) also depended on severity of patients with studies on hospitalized patients reporting from 6–73% of patients taking it. In severely ill, 36% (95% CI, 22–51%) received immunoglobulin as a therapy.

## Incubation time

Meta-analysis of 5,900 patients revealed the range of individual studies' incubation times of 4–8 days (15 studies), and a considerable heterogeneity between, and among subgroups of studies. The longest time was reported in a single 'initially asymptomatic' study (8 days; 95% CI 7.6–8.4) and the shortest for the 'all-comer' studies (4–6 days; 4 studies). In severely ill patients, the mean incubation time for one study was 4.0 days (95% CI 3.5–4.49), whereas the other two on hospitalized patients reported longer means of 7.4 and 8 days.

## Discussion

This systematic review and series of meta-analyses aims to inform COVID-19 evidence-based clinical practice by identifying trends in clinical, laboratory and radiological characteristics, as well as outcomes across the entire peer-reviewed clinical literature up until March 22, 2020. Our risk of bias analysis indicated that by normal standards, the clinical evidence on COVID-

19 that emerges from these early studies is of limited quality. This means that clinicians, researchers and policymakers must exercise caution in interpreting results from these studies. However, considering the global emergency that the spread of COVID-19 presents, there is a need to identify potential trends from the available data early in the course of the pandemic, as long as these data are analyzed using appropriate methods and over-interpretation is avoided. We considered a wide range of variables and identified a number of features associated with different severities of disease that have relevance for clinical practice. Here we also discuss the areas where evidence is lacking, to inform future research.

## Features associated with more severe disease states

Our analysis identified several factors associated with worsening clinical status in adults with COVID-19: fever, cough, dyspnea, pneumonia, any pathological CT findings, any GGO, lymphocytopenia, elevated CRP, elevated ALT and AST, increasing age and male sex. Elevated WBC count showed a strong trend for increasing prevalence with more severe disease states in moderate/severely ill patients. Elevated lymphocytes, low platelets, IL-6, erythrocyte sedimentation rate (ESR) and D-dimers showed borderline associations with worsening clinical status, but because of lack of studies, some analyses were underpowered. Fatigue, gastrointestinal symptoms, consolidation and septal thickening in CT imaging showed non-linear association patterns, suggesting that they do not have high value in clinical assessment of severity. Headache and sore throat were associated with the presence of disease, but not with worsening clinical status.

Lymphocytopenia may be indicative of stronger activation of the immune system even in early stages of COVID, eventually leading to more severe disease and has been suggested as a predictor of poor outcome in COVID-19 [55]. The virus may directly induce T-lymphocyte apoptosis through activating intrinsic and extrinsic mechanisms, e.g. via the ACE2 receptor on lymphocytes [56], or through pro-inflammatory cytokines that negatively influence lymphocyte function [57, 58]. Other factors, such as acidosis, might also contribute. However, a conclusive pathophysiological explanation regarding replication of the SARS-CoV-2 inside T-lymphocytes or viral induced-apoptosis is not clear at this point [59]. Our results are also in line with a recent meta-analysis showing raised WBC and lymphocytopenia in COVID-19 patients [60], and overproduction of cytokines, such as IL-6, correlating with disease severity [61]. Our data on lymphocytopenia, WBC and IL-6 support the hypothesis that severe cytokine release syndrome ('cytokine storm') causes more profound respiratory disease in COVID-19, which is already established in ARDS generally [62–64]. Cytokine release syndrome may be linked to increased lung injury and T-cell dysfunction [61]. Besides a natural inflammatory response to infection (typical of SARS-CoV-2), this elevated WBC count might be associated with pharmacological interventions, such as corticoids, epinephrine and beta-agonists. However, studies retrieved in this review did not specify medications used in patients with severe disease. The clear trend for elevated AST associated with more severe disease accords with other commentary on liver injury in COVID-19 [65]. Therefore, liver function should be closely monitored in patients with COVID-19 and hepatotoxic drugs should be used with caution.

The most common radiological and CT features found in patients with SARS-CoV-2 pneumonia were GGO, septal thickening and consolidation. This is similar to observations reported in other types of viral pneumonia, including those associated with SARS and MERS [66, 67]. GGO was strongly associated with severe disease. In COVID-19 patients, middle, lower and posterior lobes, peripherally and symmetrically, are most commonly affected [68]. In SARS, GGO is also described predominantly in these regions [69], suggesting that this pattern of

GGO might be a hallmark for coronavirus-related pathologies. Among studies of patients admitted to hospital we found a pattern of decreasing probability of GGO with increasing severity of illness which occurred separately in the mild/moderate (from ~90% to 40%), and moderate/severe (from 98% to 67%) groups. This might be due to the natural evolution of the disease, as consolidation represents a later stage of pulmonary injury. Therefore, the patients included in our analysis may have already been in a late phase of the disease when radiological imaging was performed, or this might reflect fast disease progression in these patients.

Fever, cough and dyspnea were the symptoms most strongly associated with more severe disease states. This trend was clear for dyspnea and fever even across the studies included in the subgroup analysis of severe patients, suggesting that these are rapidly evolving symptoms with predictive potential in individual treatment plans. Among lab variables, white blood cell count and IL-6 show particular promise as markers of severe disease. Elevated white blood cell count showed a striking trend for increasing prevalence with more severe clinical status among hospitalized patients with moderate/severe disease. Although IL-6 was only measured in 6 studies, and therefore more data are needed to increase certainty, an association with severity was clear. Most patients with severe disease had elevated CRP, but this was also common among patients with milder disease. Hence CRP appears to have utility as a marker of clinically manifesting disease, but not related to severity. PCT did not show a clear association with the dynamics of COVID-19. Slightly higher rates of patients with elevated PCT in the severely ill subgroup are likely due to higher rates of bacterial superinfection. This disconnection between PCT and COVID-19 suggests that PCT may have value as a marker of bacterial superinfection to guide rational use of antibiotics in patients with COVID-19.

## Features associated with early and mild disease

In contrast to severe disease, mild disease appears to be more common in women than men. There were more female than male patients in the two 'initially asymptomatic' studies (67% and 60% female), and overall the 11 'early' studies included higher percentages of women than men, although high heterogeneity limits confidence in these findings. Potential protective factors for women include immune modulation by sex-specific steroids and the estrogen receptor [70, 71]. The pediatric literature indicates that children are generally unlikely to develop severe disease and suggests no significant differences based on sex [72–74].

We observed a tendency for increasing prevalence of fever and cough with worsening clinical status, with frequencies as low as 13% and 8%, respectively, among studies in the 'initially asymptomatic' and 'early' study subgroups. Dyspnea was similarly infrequent in these studies. This supports the idea that these symptoms cannot be relied upon to identify early or mild disease. Headache, sore throat and GI symptoms were apparent in the 'early' subgroup in up to 20% of patients, showing association with disease presence but not with worsening clinical status. However, these symptoms are common in other rhino-, echo- and coronaviruses [75], and so do not appear to be helpful in differential diagnosis of COVID-19.

Lymphocytopenia and elevated CRP were the most common abnormalities in laboratory markers in early and mild disease, both present in at least 17% of patients in studies of initially asymptomatic individuals. Among 'early stage' studies, 20–60% of patients had lymphocytopenia, suggesting that this may be an early sign of SARS-CoV-2 infection. Pulmonary abnormalities on CT imaging were common across all patient groups. CT changes were present in at least 50% of patients in 'initially asymptomatic' studies and 64% of patients in 'early' studies. This indicates the potential of CT imaging as an early recognition tool of COVID-19. Overall, our analysis indicates that symptoms in early and mild disease are non-specific and highly heterogeneous, underlining the need for diagnostic testing to confirm infection. Mild or atypical

symptoms should not exclude suspicion of COVID-19. In such groups, CT imaging might be helpful in assessing patients for signs of COVID-19 if PCR testing is not available or negative despite high clinical suspicion.

## Gaps identified in the evidence base

Evidence to support therapeutic approaches is clearly lacking. We found only one randomized controlled study of a pharmacological treatment. As of May 3, 2020, there were 548 ongoing studies registered in ClinicalTrials.gov. Of those, 303 were clinical trials and 6 have already been withdrawn. Clinicians should be mindful of the risks of prescribing untested treatments. Evidence from these ongoing clinical trials and patient registries are clearly of critical importance. More longitudinal studies are needed to help to assess risk factors for severe disease and for poor outcomes. Specifically, the available clinical studies were of inappropriate design or of insufficient quality to identify comorbidities associated with severe disease and explore whether factors such as length of hospitalization are linked to laboratory measures or clinical outcomes.

Four studies (n = 103) described pregnant patients. Compared with other subgroups, pregnant patients may be asymptomatic more often, with potentially lower prevalence of fever compared to other subgroups. One possible explanation is that most studies evaluating pregnant women identified women who presented to hospital to give birth and were diagnosed with SARS-CoV-2 infection upon routine testing. However, the temporary immune suppression seen in pregnancy may play a role in COVID-19 pathogenesis and there are other sparse reports of potential issues in pregnancy. A recent systematic review of COVID-19 in pregnancy found that 15 of 32 women gave birth prematurely [76]. A study of 10 neonates born to pregnant mothers with confirmed SARS-CoV-2 infection, revealed several adverse outcomes such as premature labor, fetal distress, respiratory distress and even death, but there were no cases of vertical transmission of the infection [77]. Overall there is a paucity of evidence on COVID-19 in pregnant women and further research is warranted.

We identified a paucity of data on anosmia/hyposmia. Data were collected from only two studies (n = 79), and the prevalence was 0% in both. With the media attention given to this potential symptom, the lack of evidence is notable. Case reports should provide the widest possible range of symptoms since unique features of COVID-19 presentation may be poorly characterized. For laboratory variables, there were too few studies or too little data to indicate any association with COVID-19 or worsening clinical status for neutrophil levels, procalcitonin (PCT) or troponin. The level of evidence is therefore low and further studies are needed. Markers of myocardial damage such as troponin appear to be of particular interest because of reports of cardiac involvement in COVID-19, potentially related to the ACE2 receptor [78, 79].

We were unable to provide strong estimates of mortality or rate of asymptomatic disease from the available clinical data analyzed due to inappropriate study design of included studies to assess this outcome. Case fatality rates derived from 10 studies in the 'early', pregnant and 'all-comer' subgroups ranged from 2% to 2.6%. These are likely to represent the closest estimates from our analysis of the true case fatality ratio in the general adult population or slightly overestimate it, and are similar to some modelling estimates based on national statistics [80, 81] and data from the Diamond Princess cruise ship [82]. Since asymptomatic cases and patients who do not seek medical care are often not tested, and many countries have limited testing resources, the infection fatality rate is likely to be much lower. Furthermore, discharged patients may later become positive for SARS-CoV-2 and remain asymptomatic [83], causing challenges for infection control. Larger, population-based studies would provide stronger evidence for case-fatality rates and are therefore urgently needed.

## Strengths and limitations

An important strength of our review lies in its rapid assimilation of available data, just weeks after publication of some of the primary studies. We also adhered to established strict and systematic methodological criteria, including a clearly defined search strategy, public and transparent protocol and rigorous analysis techniques such as determination of risk of bias.

The current analysis builds on previous work in an initial scoping review [5]. Updating systematic reviews is generally considered to be more efficient than addressing the same question with a fresh protocol [84], and this approach is particularly appropriate in the context of new data emerging on a daily basis. We identified 72 new studies for inclusion in this systematic review published after our previous review, between February 24, 2020 and March 22, 2020. Although our methodology is broadly in line with the previous work, this large number of new studies allowed us to exclude the smallest studies (those with less than 10 patients) and therefore increase confidence in the results. In contrast to our first scoping review, this new systematic analysis used a quality effects model instead of a random effects model in order to better understand certainty of the signals based on quality and risk of bias of these studies. The quality effects model outperforms random effects where there is considerable heterogeneity.

Our conclusions are constrained by the general low quality of the available studies at this early stage of the pandemic. Studies tended to report the prevalence of symptoms at certain time points in the disease course, for example, in individuals identified through contact tracing, at hospital admission, at the onset of severe disease, or symptoms reported during hospitalization regardless of the time point. Very short follow-up periods, if any, were used. Included studies covered a wide range of populations from many countries. There are general concerns about the inclusion of COVID-19 patients in more than one publication [85]. We addressed this and sought to mitigate it by close attention to overlapping time frames, study settings and locations. Our screening process uncovered 19 overlapping studies, which were excluded from the analysis. However, we may still have failed to identify some overlapping patients, particularly from studies in the very early phase of reporting the pandemic. Although some studies used consistent methodologies, there may have been constraints on certain healthcare systems that would result in heterogeneous reporting (e.g. a shortage of tests). Our subgroup analyses aimed to mitigate this limitation as much as possible. One potential limitation of our analysis is that we did not consider a specific, well-defined question for analysis. Rather we chose a broad approach because we judged that at this early stage in the pandemic a search for signals across all the clinical data was necessary, in order to identify areas for future targeted research.

## Conclusions

In conclusion, we found worsening clinical status in COVID-19 to be associated with fever, cough, dyspnea, pneumonia, any CT findings, any GGO, lymphocytopenia, elevated WBC, elevated CRP, elevated ALT, elevated AST, increased age and male sex. Headache and sore throat were associated with the presence of disease, but not with more severe disease. Symptoms in early and mild disease are non-specific and highly heterogeneous. Although pulmonary abnormalities appear to be common even in mild disease, negative CT imaging cannot exclude suspicion of infection. There are urgent evidence gaps in terms of efficacy of treatments, characteristics of disease in severely ill patients, data from longitudinal studies, and evidence is lacking on the prevalence of anosmia and laboratory markers such as troponin, neutrophils and PCT. Caution must be exercised in interpreting the clinical data on COVID-19 because most studies published in this early stage of the pandemic have a high risk of bias and the overall quality of evidence is low. However, the emergency situation demands

decision-making based on the available evidence, and hence we offer an analysis of the entirety of the evidence base to date through our broad meta-analysis approach.

## Supporting information

**S1 File.**
(DOCX)

## Acknowledgments

We thank Maria Bjorklund (Cochrane Sweden) for designing the search strategy used in this review. We thank Dr. Clarice Lee and Dr. Laura Martins Goncalves for assisting the group with the translation services from Korean to English, Pavel Černy and Roger Crosthwaite for guiding the team supervisor (IJBN) on human resources management. This group was orga-nised via the "Cochrane TaskExchange" platform. We thank Anneliese Arno (Covidence, Aus-tralia), for providing support with the screening platform used. Hereby we declare that this study was developed by a number of international researchers associated with the Interna-tional Network of Coronavirus Disease 2019, leaded by Israel Júnior Borges do Nascimento, from the University Hospital and School of Medicine at Universidade Federal de Minas Gerais, Brazil and Medical College of Wisconsin, United States of America. In addition, the following list of medical researchers were also enrolled to the present study: Milena Soriano Marcolino (University Hospital and School of Medicine at Universidade Federal de Minas Gerais, Brazil), Silvana Mangeon Meirelles Guimarães (University Hospital and School of Medicine at Univer-sidade Federal de Minas Gerais, Brazil), Vinicius Tassoni Civile, Alvaro Nagib Attalah (Uni-versidade Federal de São Paulo, Brazil), Thilo von Groote (University of Münster, Germany), Henning Klapproth (University of Münster, Germany), Hebatullah Mohamed Abdulazeem (Universität München, Germany), Ishanka Weerasekara (The University of Newcastle, Austra-lia), Nensi Cacic, Ana Jeroncic, Tina Poklepovic Pericic, Irena Zakarija-Grkovic, Ana Marusic, Livia Puljak (University of Split School of Medicine, Croatia), Nicola Robert Bragazzi (York University, Canada), Umesh Jayarajah (University of Colombo, Sri Lanka), Donal P. O'Mathúna (The Ohio State University, United States of America), Maria Bjorklund (Lund University, Sweden), Catherine Henderson (Swanscoe Communications, United Kingdom), Abhijna Vithal Yergolkar (Ramaiah University of Applied Sciences, India), Joanna Przeźd-ziecka-Dołyk (Medical University of Wroclaw, Poland), Santino Filoso (Yorkville University, Canada), Cristina Riboni (San Matteo Hospital, Italy), Katiane Cunha (Universidade Estadual do Pará, Brazil), Ingrid Ellen Herculano dos Santos (Universidade Federal de Campina Grande, Brazil). All future communications regarding the study, as well as any requests related to the topic should be directed to the group leader (Israel Júnior Borges do Nascimento) using the following electronic address: israeljbn@ufmg.br.

## Author Contributions

**Conceptualization:** Israel Júnior Borges do Nascimento.

**Data curation:** Israel Júnior Borges do Nascimento, Thilo Caspar von Groote, Dónal P. O'Mathúna, Hebatullah Mohamed Abdulazeem, Umesh Jayarajah, Ishanka Weerasekara, Tina Poklepovic Pericic, Henning Edgar Gerald Klapproth, Livia Puljak, Nensi Cacic, Irena Zakarija-Grkovic, Silvana Mangeon Meirelles Guimarães, Alvaro Nagib Atallah, Nicola Luigi Bragazzi, Milena Soriano Marcolino, Ana Marusic, Ana Jeroncic.

**Formal analysis:** Israel Júnior Borges do Nascimento, Thilo Caspar von Groote, Dónal P. O'Mathúna, Hebatullah Mohamed Abdulazeem, Catherine Henderson, Umesh Jayarajah, Ishanka Weerasekara, Tina Poklepovic Pericic, Henning Edgar Gerald Klapproth, Livia Puljak, Nensi Cacic, Irena Zakarija-Grkovic, Silvana Mangeon Meirelles Guimarães, Alvaro Nagib Atallah, Nicola Luigi Bragazzi, Milena Soriano Marcolino, Ana Marusic, Ana Jeroncic.

**Investigation:** Israel Júnior Borges do Nascimento, Thilo Caspar von Groote, Hebatullah Mohamed Abdulazeem, Umesh Jayarajah, Ishanka Weerasekara, Tina Poklepovic Pericic, Henning Edgar Gerald Klapproth, Irena Zakarija-Grkovic, Alvaro Nagib Atallah, Milena Soriano Marcolino, Ana Marusic.

**Methodology:** Israel Júnior Borges do Nascimento, Thilo Caspar von Groote, Hebatullah Mohamed Abdulazeem, Umesh Jayarajah, Ishanka Weerasekara, Tina Poklepovic Pericic, Henning Edgar Gerald Klapproth, Livia Puljak, Irena Zakarija-Grkovic, Alvaro Nagib Atallah, Nicola Luigi Bragazzi, Milena Soriano Marcolino.

**Project administration:** Israel Júnior Borges do Nascimento.

**Resources:** Israel Júnior Borges do Nascimento, Milena Soriano Marcolino.

**Software:** Israel Júnior Borges do Nascimento.

**Supervision:** Israel Júnior Borges do Nascimento, Thilo Caspar von Groote, Catherine Henderson, Livia Puljak, Ana Jeroncic.

**Visualization:** Israel Júnior Borges do Nascimento.

**Writing – original draft:** Israel Júnior Borges do Nascimento, Thilo Caspar von Groote, Dónal P. O'Mathúna, Hebatullah Mohamed Abdulazeem, Catherine Henderson, Nicola Luigi Bragazzi, Ana Jeroncic.

**Writing – review & editing:** Israel Júnior Borges do Nascimento, Thilo Caspar von Groote, Dónal P. O'Mathúna, Hebatullah Mohamed Abdulazeem, Catherine Henderson, Tina Poklepovic Pericic, Livia Puljak, Irena Zakarija-Grkovic, Silvana Mangeon Meirelles Guimarães, Alvaro Nagib Atallah, Milena Soriano Marcolino, Ana Marusic, Ana Jeroncic.

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
