## [Decision Letter · Decision Letter 0]

15 Jul 2020

PONE-D-20-13790

Clinical, laboratory and radiological characteristics and outcomes of novel coronavirus (SARS-CoV-2) infection in humans: A systematic review and series of meta-analyses

PLOS ONE

Dear Dr. Borges do Nascimento,

Thank you for submitting your manuscript to PLOS ONE. After careful consideration, we feel that it has merit but does not fully meet PLOS ONE’s publication criteria as it currently stands. Therefore, we invite you to submit a revised version of the manuscript that addresses the points raised during the review process.

I would appreciate if you pay careful attention in your revised manuscript to the reviewer's comments.

We look forward to receiving your revised manuscript.

Kind regards,

Ehab Farag, MD FRCA FASA

Academic Editor

PLOS ONE

Journal Requirements:

"I have read the journal's policy and the authors of this manuscript have the following potential competing interests: CH has previously provided medical writing support for projects funded by Amgen, AstraZeneca, Novartis, Sobi and Takeda, unrelated to the current work. All other authors declare that no competing interests exist."

We note that one or more of the authors are employed by a commercial company: Swanscoe Communications Ltd.

2.1. Please provide an amended Funding Statement declaring this commercial affiliation, as well as a statement regarding the Role of Funders in your study. If the funding organization did not play a role in the study design, data collection and analysis, decision to publish, or preparation of the manuscript and only provided financial support in the form of authors' salaries and/or research materials, please review your statements relating to the author contributions, and ensure you have specifically and accurately indicated the role(s) that these authors had in your study. You can update author roles in the Author Contributions section of the online submission form.

2.2. Please also provide an updated Competing Interests Statement declaring this commercial affiliation along with any other relevant declarations relating to employment, consultancy, patents, products in development, or marketed products, etc. 

3. One of the noted authors is a group or consortium [International Task Force Network of Coronavirus Disease 2019 (InterNetCOVID-19)]. In addition to naming the author group, please list the individual authors and affiliations within this group in the acknowledgments section of your manuscript. Please also indicate clearly a lead author for this group along with a contact email address.

Reviewers' comments:

Reviewer's Responses to Questions

**Comments to the Author**

1. Is the manuscript technically sound, and do the data support the conclusions?

Reviewer #1: Yes

2. Has the statistical analysis been performed appropriately and rigorously? 

Reviewer #1: Yes

3. Have the authors made all data underlying the findings in their manuscript fully available?

Reviewer #1: Yes

4. Is the manuscript presented in an intelligible fashion and written in standard English?

Reviewer #1: Yes

5. Review Comments to the Author

Reviewer #1: Thank you for the opportunity to review this work. I have several comments on this meta-analysis. In general, I agree with the idea of the analysis in that there is a need to identify and establish clarity on a lot of key outcome issues with the COVID 19 pandemic.

In general, this is good work, and a very comprehensive analysis that has been done using appropriate methodology for a meta analysis. The linkage of disease severity with age, sex, radiology and labs is important and is still being used to guide prognostication and therapy.

Some issues that I would like the authors to answer:

1. Date range chosen 01-01-2019 and 03-22-2020. Considering 01-01-2019 ——> was there any coronavirus cases anywhere in the world at that time? To my best knowledge, the earliest diagnosed cases where November, 2019 in Wuhan, China. What is the objective of going back all the way to Jan,2019? Can the authors actually show us if they identified any published work from before November, 2019-Jan 2019?

2. I also remain concerned about the very inclusive “all types of research’ accepted in inclusion criteria. One problem with COVID19 research and or data is the lack of value from sample size datasets. We have seen along the evolution of science associated with the disease that small datasets have fueled speculation and added nothing to actual evidence and sometimes when prospective observational n=10 studies have shown ‘encouraging’ outcomes, these have not been validated by larger studies especially the larger studies with adequate controls. Did the authors consider breaking down their results by types of studies? I dare say that there were not many RCTs in the picture in March 2020 when this work was done. Therefore a lot of the included studies here are case series. While I agree that the initial impetus from Wuhan was to publish a lot of case series, now things are very different.

3. I would strongly encourage the authors to make their work more concise and crisp. I know that this is a series of several meta-analysis put together, but it is exhausting for an average reader to get through this paper and take any clinical benefit from it, since the messages are lost in a lot of textual information.

4. Finally, the evidence of science and related outcomes is constantly progressing in COVID19. Would the authors want to add relevant information for the reader to bring this work up to the mark with what is happening now? Would the addition of new studies that have been published from March till date make a big difference in their outcomes?

6. PLOS authors have the option to publish the peer review history of their article (what does this mean?). If published, this will include your full peer review and any attached files.

Reviewer #1: **Yes: **Ashish K Khanna

---

## [Editor Report · Decision Letter 1]

2 Sep 2020

Clinical, laboratory and radiological characteristics and outcomes of novel coronavirus (SARS-CoV-2) infection in humans: A systematic review and series of meta-analyses

PONE-D-20-13790R1

Dear Dr. Israel Junior Borges do Nascimento

We’re pleased to inform you that your manuscript has been judged scientifically suitable for publication and will be formally accepted for publication once it meets all outstanding technical requirements.

Kind regards,

Ehab Farag, MD FRCA FASA

Academic Editor

PLOS ONE
---

## [Editor Report · Acceptance letter]

4 Sep 2020

PONE-D-20-13790R1 

Clinical, laboratory and radiological characteristics and outcomes of novel coronavirus (SARS-CoV-2) infection in humans: A systematic review and series of meta-analyses 

Dear Dr. Borges do Nascimento:

I'm pleased to inform you that your manuscript has been deemed suitable for publication in PLOS ONE. Congratulations! Your manuscript is now with our production department. 

Kind regards, 

on behalf of

Dr. Ehab Farag 

Academic Editor

PLOS ONE